# INFERENCE SCALING LAWS:
# AN EMPIRICAL ANALYSIS OF COMPUTE-OPTIMAL INFERENCE FOR LLM PROBLEM-SOLVING

**Yangzhen Wu**[1][*]**, Zhiqing Sun**[2]**, Shanda Li**[2]**, Sean Welleck**[2]**, Yiming Yang**[2]

[1]Institute for Interdisciplinary Information Sciences, Tsinghua University
[2]School of Computer Science, Carnegie Mellon University
wuyangch21@mails.tsinghua.edu.cn
{zhiqings, shandal, swelleck, yiming}@cs.cmu.edu
https://thu-wyz.github.io/inference-scaling/

## ABSTRACT

While the scaling laws of large language models (LLMs) training have been extensively studied, optimal inference configurations of LLMs remain underexplored. We study *inference scaling laws* (*aka* test-time scaling laws) and *compute-optimal inference*, focusing on the trade-offs between model sizes and generating additional tokens with different inference strategies. As a first step towards understanding and designing compute-optimal inference methods, we studied cost-performance trade-offs for inference strategies such as greedy search, majority voting, best-of-$n$, weighted voting, and two different tree search algorithms, using different model sizes and compute budgets. Our findings suggest that scaling inference compute with inference strategies can be more computationally efficient than scaling model parameters. Additionally, smaller models combined with advanced inference algorithms offer Pareto-optimal trade-offs in cost and performance. For example, the Llemma-7B model, when paired with our novel tree search algorithm, consistently outperforms the Llemma-34B model across all tested inference strategies on the MATH benchmark. We hope these insights contribute to a deeper understanding of inference scaling laws (test-time scaling laws) for LLMs.

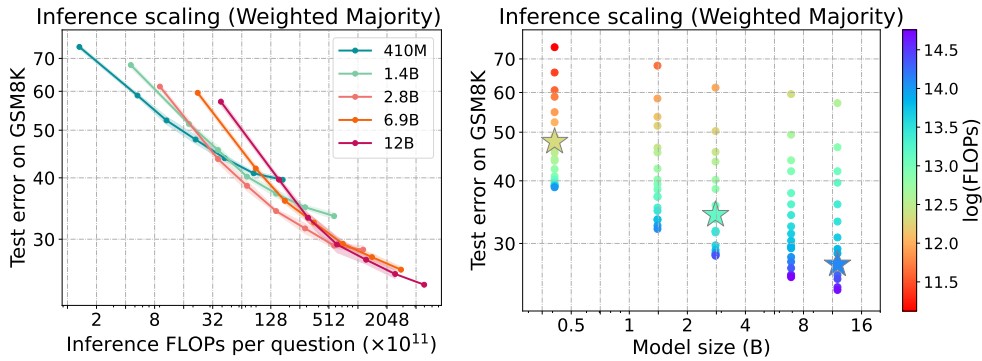

Figure 1: **Inference scaling laws** exhibited for Pythia (Biderman et al., 2023) models and **GSM8K** test error. We evaluate the error rate (lower is better) of models using various sizes and numbers of sampled solutions for weighted majority voting. *Left:* the error rate for each model size decreases steadily as inference-compute increases, and converges at the end. *Right:* the optimal model size (shown as stars for $2^{41}$, $2^{44}$, and $2^{47}$ FLOPs) varies based on the inference-time compute budget. For instance, smaller models are compute-optimal at $2^{41}$ and $2^{44}$ FLOPs. Both axes are $\log$ scale.

---

[*]Work done during the visit at Carnegie Mellon University

# 1 INTRODUCTION

Scaling laws of neural networks (Hestness et al., 2017; Rosenfeld et al., 2020) have been established across a range of domains, including language modeling (Kaplan et al., 2020; Hoffmann et al., 2022; OpenAI, 2023), image modeling (Henighan et al., 2020; Yu et al., 2022; Peebles & Xie, 2023), video modeling (Brooks et al., 2024), reward modeling (Gao et al., 2023), and board games (Jones, 2021). These studies have demonstrated how model performance is influenced by both the size of the model and the amount of training compute. However, there is limited knowledge on how varying the compute during *inference* affects model performance after the model has been trained.

To improve the task performance of large language models (LLMs), inference techniques typically involve additional compute as a *performance maximization* step at inference time (Nye et al., 2021; Wei et al., 2022; Wang et al., 2023b; Yao et al., 2023; Chen et al., 2024b). The computational cost of these techniques must be taken into account for *compute-optimal inference*. For example, Monte Carlo Tree Search (MCTS) may improve task performance, but it potentially requires much more compute than simply sampling solutions multiple times (Jones, 2021). Generally speaking, we need a comprehensive understanding of how various inference-time methods (e.g., best-of-$n$, majority voting (Wang et al., 2023a; Li et al., 2023)) trade off between performance and cost. To improve our understanding, this paper presents a thorough empirical evaluation with careful analysis over various configurations of representative LLMs and inference algorithms.

Specifically, we explore how to select an optimal size for the language model and an effective inference strategy (e.g., greedy search, majority voting, best-of-$n$, weighted voting, and their tree-search variants) to maximize performance (i.e., accuracy) with a given compute budget. We control the inference compute (FLOPs) of a fixed model by generating more tokens through the language model[1], sampling further candidate solutions, and ranking them with a reward model. We analyze the performance of fine-tuned models of various sizes given different inference FLOPs on mathematical reasoning benchmarks (e.g., GSM8K test set (Cobbe et al., 2021) and MATH500 test set (Hendrycks et al., 2021b; Lightman et al., 2024)). Our experiments cover several model families, including general-purpose LLMs (e.g., Pythia (Biderman et al., 2023) & Mistral (Jiang et al., 2023)) as well as math-specialized ones (e.g., Llemma (Azerbayev et al., 2024)).

Our results on Pythia (Fig. 1) illustrate how performance scales with increased inference compute across various model sizes. Typically, increasing the compute budget leads to higher accuracy until the accuracy reaches saturation. As the compute budget increases, smaller models initially perform better than larger ones, but once the accuracy of the smaller models saturates, the larger models have favorable performance. The right panel of Figure 1 demonstrates that the optimal model size for inference varies with different levels of computational budgets. However, in real-world deployment, the available compute is typically much lower than the point where the accuracy of relatively small models saturates and larger models begin to show their advantage (as shown in Fig. 4, where the 7B model outperforms the 34B model until 128 Llemma 7B solutions are sampled). This indicates that relatively smaller models could be more compute-optimal for inference.

We analyze the asymptotic behavior of sampling and voting-based inference strategies, showing their convergence upper bound and rate of convergence. Given a dataset, the accuracy of the language model will ultimately saturate to a fixed limit which is determined by the output probabilities assigned by the model, exhibiting exponential convergence speed through sampling and voting. This implies that, without an oracle verifier, simple strategies like sampling cannot achieve perfect accuracy even with an infinite number of samples, leading to diminishing returns. Therefore, this highlights the necessity for more sophisticated inference algorithms.

We have also found that the commonly-used MCTS method does not perform well with weighted voting, as it often yields many unfinished solutions, hence having less effective votes. To address this issue, we propose a novel tree search algorithm, *REward BAlanced SEarch* (REBASE), which pairs well with weighted voting and achieves a Pareto-optimal trade-off between accuracy and inference compute. The key idea of REBASE is to use a node-quality reward to control node expansion, which eliminates the need for explicit rollouts while ensuring enough candidate solutions for voting.

---

[1] Following Uesato et al. (2022), we refer to the main language model generating outputs as the *policy* model. It can be paired with a *reward* model, which scores outputs from the policy model to facilitate inference.

In our experiments, REBASE consistently outperforms sampling and MCTS methods across all settings, models, and tasks. Importantly, we find that REBASE with a *smaller* language model typically achieves a Pareto-optimal trade-off. For instance, we show that the Llemma-7B model can achieve competitive accuracy to a Llemma-34B model while using $2\times$ less FLOPs when evaluating on MATH500 (Fig. 4) or GSM8K (Fig. 5). Moreover, Llemma-7B with REBASE outperforms Llemma-34B with standard majority voting across *all* compute budgets. Our results show the value of using smaller models with advanced inference-time algorithms, and the benefits of new algorithms for achieving better returns on inference-time compute.

Our contributions are summarized as follows:

- We explore new inference scaling laws and compute-optimal inference by evaluating the performance of various model sizes under a fixed inference strategy. We show that smaller models can outperform larger ones under the same compute budget by increasing the number of samples.

- We provide new theoretical analysis of the scaling behavior of voting methods, presenting convergence bounds and rates. Our analysis shows performance limits and diminishing returns from sampling, pointing to the need for more sophisticated inference algorithms.

- We formulate a new compute-optimal inference problem and propose a novel tree search algorithm, REBASE, which is compute-optimal compared to widely-used sampling and MCTS methods. Our results show benefits of using smaller models with advanced inference algorithms, and new algorithms for achieving better cost-performance tradeoffs.

## 2  RELATED WORKS

**Scaling laws.**  Recent research on scaling laws has established that model performance follows predictable power-law relationships with respect to the number of parameters, the size of the training dataset, and the available compute (Hestness et al., 2017; Rosenfeld et al., 2020). The seminal work by Kaplan et al. (2020) demonstrates that the test loss of language models decays as a function of model size and data in a highly regular manner. Subsequent studies refine these initial observations and extend them into more diverse settings (Hoffmann et al., 2022; Alabdulmohsin et al., 2022; Muennighoff et al., 2023; Lin et al., 2024; Goyal et al., 2024b). However, most of these existing works are primarily focused on the *training* regime. Sardana et al. (2024) study scaling laws taking both training and inference into account, but only a fixed inference algorithm is considered. In comparison, our work systematically demonstrate *inference* **scaling laws**, i.e., LLM problem-solving performance improves with increased inference-time compute budget, and we propose and study compute-optimal inference.

**Inference strategies and inference-time compute utilization in LLM problem-solving.**  A variety of inference strategies have been developed to generate sequences with a trained model (Welleck et al., 2024). Deterministic methods such as greedy decoding and beam search (Teller, 2000; Graves, 2012) find highly probable sequences which typically have decent quality but lacks diversity. Sampling algorithms (e.g., temperature sampling (Ackley et al., 1985)) can produce a diverse set of results which are then aggregated to achieve higher accuracy (e.g., via the self-consistency approach (Wang et al., 2023a)). Recent methods combine search algorithms with LLMs, including breadth-first or depth-first search (Yao et al., 2023), Monte-Carlo Tree Search (MCTS) (Zhang et al., 2023; Zhou et al., 2024; Liu et al., 2024; Choi et al., 2023), and guided beam search (Xie et al., 2023). Several prior studies also find that LLM problem-solving performance can be improved by outputting "dummy" tokens at inference time (Goyal et al., 2024a; Pfau et al., 2024). All of these methods show that using search at inference time can lead to performance gains at the cost of increased inference-time compute, but they do not characterize the cost-performance trade-off systematically. We are the first to formulate and study **compute-optimal inference**, analyzing the trade-off between compute budget and problem-solving performance and proposing the REBASE method that is empirically Pareto-optimal. Concurrently, Snell et al. (2025) also study how to scale inference-compute optimally and provide complementary insights, since we consider more model families and sizes while they study several different inference strategies.

**Mathematical Reasoning with LLMs.**  Large language models have made significant progress in recent years, and have exhibited strong reasoning abilities (Brown et al., 2020; Hoffmann et al., 2022; Lewkowycz et al., 2022; Chowdhery et al., 2023). Mathematical problem solving is a key

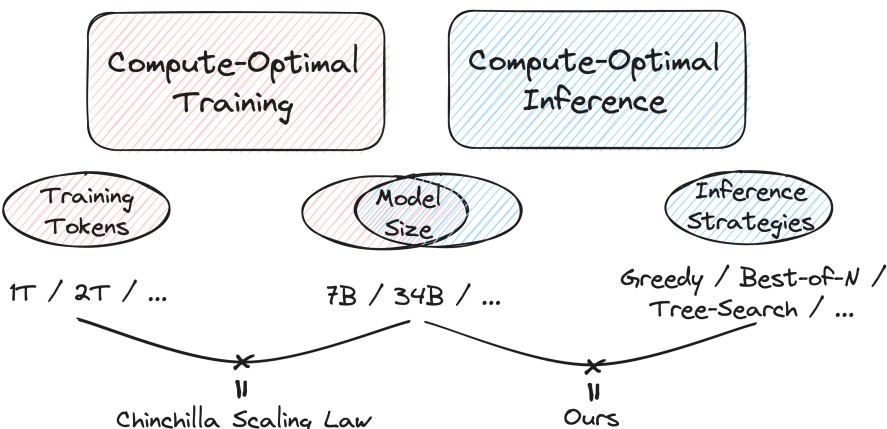

Figure 2: **Illustration of compute-optimal scaling laws in training and inference.** The Chinchilla scaling law (Hoffmann et al., 2022) shows how to choose a model size and number of training tokens under a training-compute budget, while our work shows how to choose a model size and an inference strategy under an inference-compute budget.

task for measuring LLM reasoning abilities (Cobbe et al., 2021; Hendrycks et al., 2021b). Ling et al. (2017) first developed the method of producing step by step solutions that lead to the final answer. Later, Cobbe et al. (2021) extended the work by training a verifier for evaluating and ranking solutions. Subsequent research has shown the performance benefits of inference-time techniques such as majority voting and weighted majority voting (Lewkowycz et al., 2022; Li et al., 2023). We choose problem solving in mathematics as the task to study compute-optimal strategies since it allows us to accurately evaluate problem solving ability.

## 3 COMPUTE-OPTIMAL INFERENCE FOR PROBLEM-SOLVING

We explore the following question: *Given a fixed FLOPs budget, how should one select an optimal model size for the policy model, and an effective inference strategy to maximize performance (i.e., accuracy)?* We are the first to formulate this problem and study the associated inference-time scaling laws, setting our work apart from previous scaling law studies (Fig. 2).

To address this, we represent the problem-solving error rate $E(N, T; \mathcal{S})$ as a function of the number of model parameters $N$, the number of generated tokens $T$ and the inference strategy $\mathcal{S}$. The computational budget $C$ is a deterministic function $\text{FLOPs}(N, T; \mathcal{S})$, based on $N$ and $T$. Our goal is to minimize $E$ under the test-time compute constraint $\text{FLOPs}(N, T, \mathcal{S}) = C$:

$$(N_{\text{opt}}(C), T_{\text{opt}}(C); \mathcal{S}) = \underset{(N,T,\mathcal{S}) \text{ s.t. } \text{FLOPs}(N,T,\mathcal{S})=C}{\arg\min} E(N, T; \mathcal{S})$$

where $N_{\text{opt}}(C)$ and $T_{\text{opt}}(C)$ denote the optimal allocation of a computational budget $C$.

Here, the inference compute (FLOPs) for a fixed model can be modulated by generating more tokens with the policy model and an inference strategy, e.g., sampling additional candidate solutions and subsequently ranking them using a reward model. As the inference strategies, we primarily consider sampling and tree-search approaches paired with re-ranking or majority voting. This includes greedy search, majority voting, best-of-$n$, weighted voting, and their tree-search variants.

### 3.1 INFERENCE STRATEGIES

We consider the following inference strategies which are popularly used in practice:

- **Greedy search.** This strategy generates tokens one at a time by selecting the highest probability token at each step. It is computationally efficient but often suboptimal in terms of diversity.

- **best-of-$n$.** This strategy, also known as rejection sampling, generates a set of candidates and chooses the one with the highest score given by the reward model.

- **Majority voting.** In this strategy, a set of candidates are generated, and the final answer to the problem is determined by the most frequently occurring answer in all the outputs.

- **Weighted majority voting.** This strategy is a variant of majority voting in which the candidates are weighted based on the scores given by the reward model.

We say a strategy is **sampling-based** if it uses a standard autoregressive sampling algorithm (e.g., temperature sampling) to generate the candidate set (greedy search is separate, in that it only has a single deterministic candidate). A **tree-search** variant uses a tree search to generate the candidate set. Before discussing tree-search methods, we analyze sampling-based voting below.

**Theoretical analysis of sampling-based voting.** We present theoretical results on the asymptotic behavior of voting-based strategies given infinite compute in Theorems 1 & 2. Informally, we show that the accuracy of standard/weighted majority voting converges with infinite samples, and the limit only depends on the distribution modeled by the language model (and the reward model). This theoretical finding is also aligned with our empirical findings shown in Sec. 4.2, which show saturation at high sampling budgets. The proofs are presented in Appendix A.

**Notations and assumptions.** Let $\mathcal{V}$ be a *finite* vocabulary and $\mathcal{V}^*$ its Kleene closure, i.e., the set of all strings. Given a problem $x$, we say a language model answers $y$ to this problem if the model outputs $r e y$ where $r \in \mathcal{V}^*$ can be any "reasoning path" and $\mathrm{e} \in \mathcal{V}$ denotes a special token that marks the end of reasoning. We further assume that the answer string is always shorter than $L$ tokens, i.e., $|y| \leq L$ for some fixed $L \in \mathbb{N}^*$ where $|y|$ denotes the length of $y$. For a language model $\pi$, denote by $\pi(v|w)$ the probability of generating $v$ given input (prompt) $w$. For a reward model $\rho$, denote by $\rho(v)$ the score it assigns to the string $v$. We use $\mathbb{I}$ to denote the indicator function.

**Theorem 1.** *Consider a dataset $\mathcal{D} = \{(x_i, y_i)\}_{i=1}^m$ where $x_i$ and $y_i$ denotes input and true answer, respectively. For a language model $\pi$, denote by $\mathrm{acc}_n^{\mathrm{MV}}(\mathcal{D}; \pi)$ the accuracy on $\mathcal{D}$ using majority voting with $n$ samples. Following the notations and assumptions defined above, we have:*

$$\lim_{n \to +\infty} \mathrm{acc}_n^{\mathrm{MV}}(\mathcal{D}; \pi) = \frac{1}{m} \sum_{i=1}^m \mathbb{I}\left[ y_i = \arg\max_{|y| \leq L} \sum_{r \in \mathcal{V}^*} \pi(rey|x_i) \right] \text{ (almost surely)};$$

$$\text{and} \quad \mathbb{E}\left[ \mathrm{acc}_n^{\mathrm{MV}}(\mathcal{D}; \pi) \right] = \frac{1}{m} \sum_{i=1}^m \mathbb{I}\left[ y_i = \arg\max_{|y| \leq L} \sum_{r \in \mathcal{V}^*} \pi(rey|x_i) \right] - \mathcal{O}(c^{-n})$$

*for some constant $c > 1$.*

**Theorem 2.** *Consider a dataset $\mathcal{D} = \{(x_i, y_i)\}_{i=1}^m$. For a language model $\pi$ and a reward model $\rho$, denote by $\mathrm{acc}_n^{\mathrm{WV}}(\mathcal{D}; \pi, \rho)$ the accuracy on $\mathcal{D}$ using weighted majority voting with $n$ samples. Following the notations and assumptions defined above, we have:*

$$\lim_{n \to +\infty} \mathrm{acc}_n^{\mathrm{WV}}(\mathcal{D}; \pi, \rho) = \frac{1}{m} \sum_{i=1}^m \mathbb{I}\left[ y_i = \arg\max_{|y| \leq L} \sum_{r \in \mathcal{V}^*} \pi(rey|x_i)\rho(x_irey) \right] \text{ (almost surely)};$$

$$\text{and} \quad \mathbb{E}\left[ \mathrm{acc}_n^{\mathrm{WV}}(\mathcal{D}; \pi, \rho) \right] = \frac{1}{m} \sum_{i=1}^m \mathbb{I}\left[ y_i = \arg\max_{|y| \leq L} \sum_{r \in \mathcal{V}^*} \pi(rey|x_i)\rho(x_irey) \right] - \mathcal{O}(c^{-n})$$

*for some constant $c > 1$.*

**Remarks.** Theorems 1 & 2 state the convergence of the accuracy with increasing number of samples, indicating that the performance gains of using more samples will saturate for any fixed models. The limit is determined by the likelihood of generating the correct answers through all possible reasoning paths (and the likelihood should be viewed as a weighted sum for weighted majority voting). This motivates us to consider inference algorithms that search for "good" reasoning paths, such as the tree-search-based variants detailed in Sec. 3.1.1 & 3.1.2.

Theorem 1 & 2 also present insights to compare standard majority voting with weighted majority voting. Informally, as long as the reward model is "better than random", i.e., assigning higher rewards to correct solutions on average, the accuracy limit of weighted majority voting is higher than that of majority voting. In our experiments, we consistently find that weighted majority voting dominates majority voting. Thus, we focus on best-of-$n$ and weighted majority voting in the main paper and defer majority voting results to Appendix D.

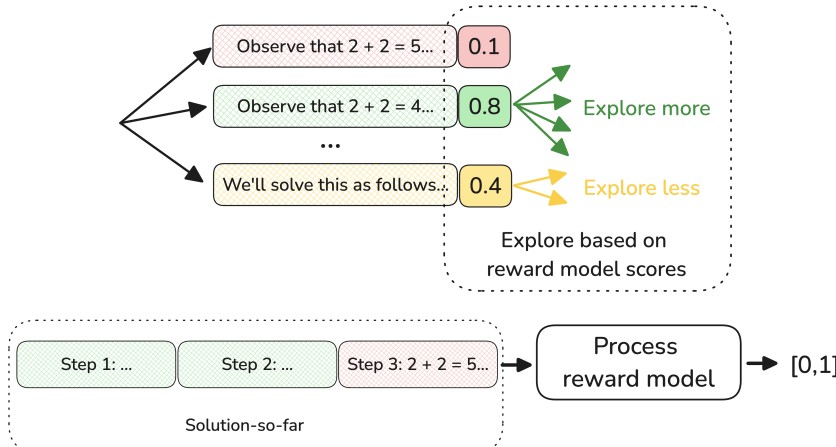

Figure 3: Illustration of one iteration of REward BAlanced SEarch (REBASE).

### 3.1.1 MONTE CARLO TREE SEARCH (MCTS)

Monte Carlo Tree Search (MCTS) has proven effective in domains such as board games where strategic decision-making is required (Silver et al., 2016; 2017; Jones, 2021). Recent work has shown that adapting MCTS to the context of LLMs can enhance the text generation process (Zhang et al., 2023; Zhou et al., 2024; Liu et al., 2024; Choi et al., 2023; Chen et al., 2024a; Tian et al., 2024; Chen et al., 2024a). In this context, MCTS is paired with a value model to score and guide the exploration steps. For additional background, we provide a review of MCTS in Appendix B.

Recent work in MCTS or its variants mainly focus on improving the performance (e.g., accuracy) on the studied tasks. However, generic comparisons of MCTS with conventional methods like best-of-$n$ and majority voting in terms of computational budget, measured in generated tokens or processing time are scarce or indicate potentially unfavorable cost-performance tradeoffs. For example, MCTS consumes substantially more resources, often requiring dozens of times more generated tokens than simpler methods. Specifically, a significant portion of the paths in the search tree are used to estimate and select nodes, and these paths do not necessarily become a part of the final candidate solution, although MCTS ensures that the sampled solutions comprise high-quality intermediate steps. In contrast, sampling methods generate multiple solutions in parallel and independently, and all the generated sequences are included in the candidate solutions. However, the intermediate steps in these sequences are not guaranteed to be of high quality, as there is no mechanism for pruning poor steps or exploiting promising ones.

This highlights the need for a new tree search method that can achieve a comparable (or better) performance as MCTS, and that is computationally less costly, with a cost similar to weighted majority voting and best-of-$n$. This motivates our new method, Reward Balanced SEarch (REBASE).

### 3.1.2 REWARD BALANCED SEARCH (REBASE)

The REBASE tree search method, illustrated in Fig. 3, inherits the exploitation and pruning properties of tree search, while using a reward model alone to estimate quality of intermediate nodes. This saves inference compute compared to methods such as MCTS, since it does not involve estimate node quality with explicit rollouts. In short, the underlying idea is to use a process reward model to determine how much each node should be expanded at each depth. Namely, REBASE expands nodes at a given depth according to their softmax-normalized reward scores, subject to a total expansion budget. We describe this procedure in more detail below.

**Notations.** We view the fine-tuned LLM as a policy $\pi_\theta$ which generates the solution step by step. Given a question $x$ and the first $k$ steps of a solution $r_1 \cdots r_k$, the $(k+1)$-th step is sampled from $\pi_\theta(\cdot | xr_1 \cdots r_k)$. REBASE generates a solution tree during inference, in which the root node is the question $x$, and other nodes corresponds to solution steps. When generating solution trees, we generate children of $r_k$ by sampling from $\pi_\theta(\cdot | xr_1 \cdots r_k)$. We use the corresponding solution step to denote a node. The reward of a node $r_k$ is generated by the PRM: $R(r_k) := R(xr_1 \cdots r_k)$.

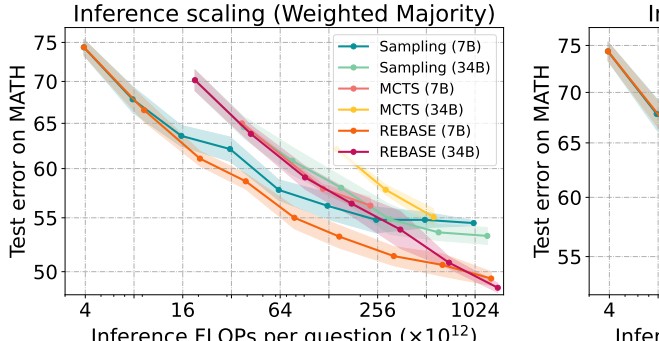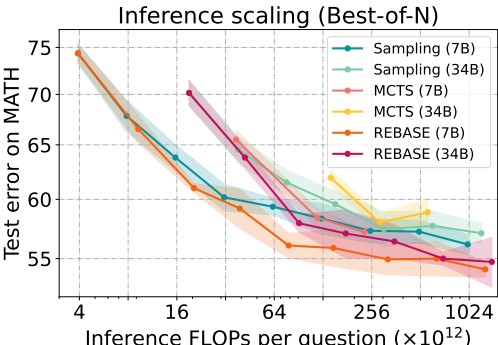

Figure 4: **MATH inference scaling across inference strategies and model sizes** (lower is better). Detailed MCTS configurations can be found in Appendix B. The left/right panel shows the error rate on MATH based on weighted majority/best-of-$n$. REBASE is the compute-optimal strategy at all budgets, with 7B typically the optimal model size.

**Initialization.** Given the question $x$, a balance temperature $T_b > 0$, and target number of generated solutions $N$, we sample $N$ instances of the first step for the question, yielding all the nodes of depth 1 in the search tree. We let the sampling budget of depth 0, $B_0$, to $N$ at initialization.

**Reward assignment and update.** In the $i$-th iteration, the PRM assigns the rewards to all the nodes at depth $i$. After that, the algorithm examines whether the solutions up to depth $i$ are complete. Supposing there are $C_i$ completed solutions, we update the sampling budget using $B_i \leftarrow B_{i-1} - C_i$. If $B_i = 0$, the process ends, and we obtain $N$ solutions.

**Exploration balancing and expansion.** For all of the nodes $n_j$ with reward $R(n_j)$ in the depth $i$ of the tree, we calculate the expansion width of the $n_j$ as:

$$W_j = \text{Round}\left( B_i \frac{\exp\left(R(n_j)/T_b\right)}{\sum_k \exp\left(R(n_k)/T_b\right)} \right). \tag{1}$$

Then we sample $W_j$ children for $n_j$ for all the nodes in depth $i$, and start the next iteration.

## 4 EXPERIMENTS

Our experiments are centered around two main questions:

- **Compute-optimal model size**: How does performance scale as inference-time compute is increased with a fixed inference strategy, but with varying model size?

- **Compute-optimal inference strategy**: How does performance scale as inference-time compute is increased with various inference strategies (and various model sizes)?

We detail our experimental setup below.

### 4.1 SETUP

**Datasets.** We conduct experiments on two mathematical problem-solving datasets to investigate the effects of scaling inference compute for both challenging and simpler problems. Specifically, MATH (Hendrycks et al., 2021a) and GSM8K (Cobbe et al., 2021) are datasets containing high school mathematics competition-level problems and grade-school level mathematical reasoning problems, respectively. Following Lightman et al. (2024); Wang et al. (2024); Sun et al. (2024), we use the MATH500 subset as our test set.

**Policy model (solution generator).** To study the how performance scales as inference compute is increased using a fixed strategy, the primary axis of variation is model size. Therefore, we choose Pythia (Biderman et al., 2023) as our base models, since various model sizes are available in the Pythia family. To study inference scaling under different inference strategies (e.g., tree search, weighted majority voting), we use math-specialized Llemma models (Azerbayev et al., 2024). We finetune these models on the MetaMath dataset (Yu et al., 2024) using full parameter supervised

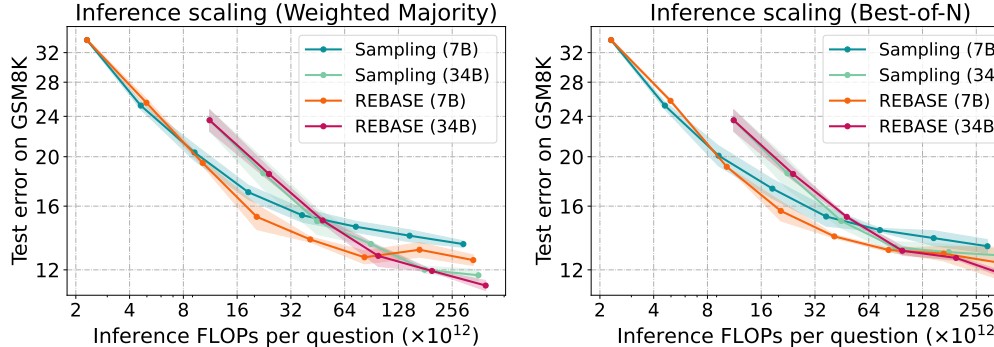

Figure 5: **GSM8k inference scaling across inference strategies and model sizes** (lower is better). The left/right panel shows the problem-solving error rate on GSM8K based on weighted majority/best-of-$n$. MCTS is not included in the comparison because of its poor compute-accuracy trade-off. REBASE is the compute-optimal inference strategy, and the optimal model size varies.

fine-tuning (Full-SFT), The finetuning configuration is given in the Appendix. Additionally, we test the Mistral-7B (Jiang et al., 2023) to expand our findings across different models and architectures.

**Reward model.** All of the experiments use the same Llemma-34B reward model, which we fine-tuned on the synthetic process reward modeling dataset, Math-Shepherd (Wang et al., 2024). We added a reward head to the model, enabling it to output a scalar reward at the end of each step.

**Inference configuration.** We use sampling and tree search methods to generate multiple candidates, and select the answer through best-of-$n$, majority voting, or weighted voting. Each configuration is run multiple times to calculate the mean and variance, which mitigates effects from randomness and thereby improves the reliability of our conclusions. Unless explicitly stated otherwise, each point in the figures in this section corresponds to $2^i$ samples, where $i$ is an integer starting from 0.

## 4.2 COMPUTE-OPTIMAL MODEL SIZE

To compare the inference compute budgets of different models, we plot the figures with the number of FLOPs used per question during inference. We compute the inference FLOPs based on the commonly-used formula proposed by Kaplan et al. (2020).

**Scaling law of compute-optimal inference for model size.** Fig. 1 shows the relationship between inference compute and error rate for different model sizes. The error rate first decreases steadily and then starts to saturate. Initially, sampling many times from smaller models is compute-optimal. At larger compute budgets the larger models are preferable, since the performance of small models has saturated. As highlighted in the right panel of Fig. 1, the optimal model size varies based on the inference budget. We performed a regression analysis on inference FLOPs $C$ and model sizes $N$ to establish a relationship between a given computational budget and its optimal model size. The resulting equation, $\log_{10}(C) = 1.19 \log_{10}(N) + 2.03$, lets us estimate the optimal inference model size for a specific compute budget.

**Llemma-7B achieves competitive accuracy to Llemma-34B with less compute.** Fig. 4 and Fig. 5 shows the relationship between error rate and inference FLOPs for Llemma 7B and Llemma 34B using different inference strategies. Llemma-7B requires around $2\times$ less total FLOPs than Llemma-34B to achieve comparable accuracy. This held across inference strategies (sampling strategies, MCTS, REBASE) and tasks (MATH, GSM8K). This result suggests that, with the same training dataset and model family, generating more tokens with a suitable inference strategy using a smaller model can have more favorable cost-performance tradeoffs than using a larger model.

## 4.3 COMPUTE-OPTIMAL INFERENCE STRATEGY

**REBASE is Pareto-optimal.** REBASE consistently achieves the best cost-performance tradeoffs, outperforming the sampling-based methods in all settings when fixing the model and the evaluation task (Fig. 4, 5, 6, and 7). For example, in Fig. 4, REBASE is the compute-optimal strategy at all

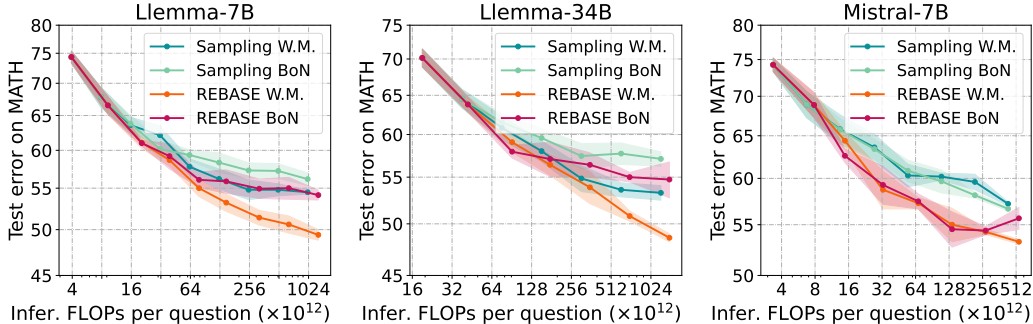

Figure 6: **MATH inference scaling across inference strategies and models** (lower is better). The tested models are Llemma-7B (left), Llemma-34B (middle), & Mistral-7B (right). In the legend, W.M. and BoN refer to weighted majority and best-of-$n$, respectively.

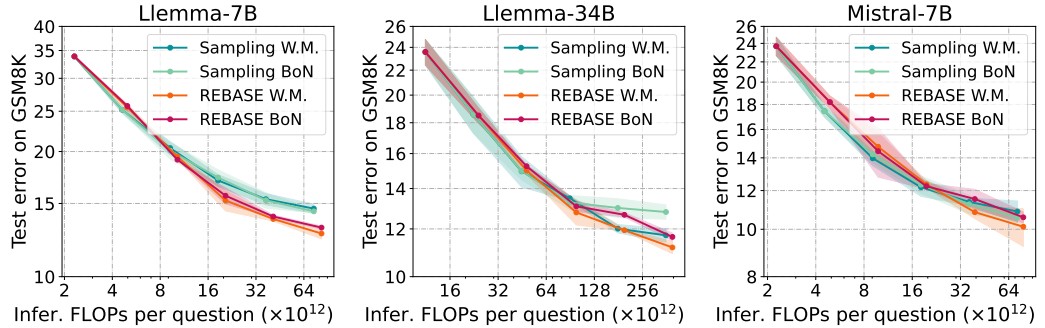

Figure 7: **GSM8K inference scaling across inference strategies and models** (lower is better). The tested models are Llemma-7B (left), Llemma-34B (middle), & Mistral-7B (right). In the legend, W.M. and BoN refer to weighted majority and best-of-$n$, respectively.

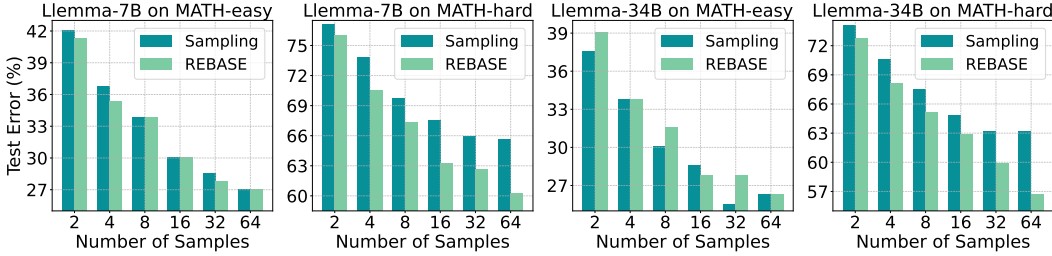

Figure 8: **Comparisons of sampling and REBASE** using weighted majority voting on MATH-easy problems (levels 1-2) and MATH-hard problems (levels 3-5). The tested models are Llemma-7B and Llemma-34B.

inference compute budgets, with 7B typically the optimal model size. On the other hand, MCTS underperforms the sampling-based methods at each compute budget, likely due to its costly rollouts (Fig. 4) compared to the efficient use of the reward model in REBASE.

Tab. 1 shows that REBASE achieves better accuracy with a lower compute budget compared to sampling-based weighted voting. With the 7B model, REBASE achieves higher accuracy with 7 times less compute. This finding is novel, and differs from previous tree search methods that typically improve the performance at the cost of higher computational expense compared to sampling-based voting (Chen et al., 2024a; Xie et al., 2023).

**REBASE yields greater gains on hard problems.** The MATH dataset assigns each problem a difficulty level from 1 to 5. This enables a finer-grained analysis of the relationship between problem

Table 1: **REBASE with a lower compute budget achieves better accuracy** compared to sampling with a higher compute budget. We use weighted voting to aggregate candidates for both sampling and REBASE.

|  | # Samples | FLOPs | MATH500 Accuracy (%) |
|---|---|---|---|
| Mistral-7B | | | |
| Sampling | 256 | $8.70 \times 10^{14}$ | 42.8 |
| REBASE | 32 | $\mathbf{1.36 \times 10^{14}}$ | **45.0** |
| Llemma-7B | | | |
| Sampling | 256 | $10.0 \times 10^{14}$ | 45.5 |
| REBASE | 32 | $\mathbf{1.48 \times 10^{14}}$ | **46.8** |
| Llemma-34B | | | |
| Sampling | 64 | $12.1 \times 10^{14}$ | 46.7 |
| REBASE | 32 | $\mathbf{7.08 \times 10^{14}}$ | **49.2** |

difficulty and inference strategy. We divide the MATH test set into MATH-easy (levels 1-2) and MATH-hard (levels 3-5) and compare REBASE to sampling on these two subsets. The results are shown in Fig. 8. The performance of sampling and REBASE on easy problems (levels 1-2) is comparable. However, REBASE demonstrates a significant advantage on harder problems (levels 3-5). This suggests that advanced inference strategies like tree search are especially effective for solving difficult problems.

**REBASE saturates later than sampling with higher accuracy.** From Fig. 6 and Fig. 7, we observe that both sampling and REBASE saturate early in GSM8K and relatively late in MATH. We attribute this to the difference of in difficulty levels between GSM8K and MATH. Specifically, the LLM may assign high probability only to correct solutions in easy problems, but spread probability mass across solutions in harder problems. Thus, harder problems may require aggregating over more solution paths to converge to the distribution over answers shown in Theorems 1 & 2. On MATH (Fig. 6), we see that REBASE finally saturates with a higher accuracy than sampling. We hypothesize the reason is that drawing samples from REBASE corresponds to sampling from a policy that assigns high probability to true answers compared to sampling from the underlying language model. If this was indeed the case, Theorems 1 & 2 indicate that the upper bound would become higher. We leave formally analyzing the behavior of tree search algorithms as interesting future work.

## 5 CONCLUSIONS

We study the relationship between task performance and the amount of compute expended during inference for various model sizes, model families, and inference strategies, to form empirical *inference scaling laws*. These relationships let us reason about *compute-optimal inference*: inference configurations that give the best performance at a given compute budget.

Our results lead to three main takeaways. First, we find that using a smaller model and generating more tokens in an inference strategy often outperforms using a larger model at a fixed compute budget. This has implications for models deployed in the real world, where inference compute is constrained in various ways. Specifically, it is potentially beneficial to deploy smaller models with more sophisticated inference strategies for better cost-performance trade-off. Second, we show that in the limit of infinite compute (allocated by drawing more samples), sampling-based majority voting strategies inevitably saturate to a distribution that depends on the underlying generation policy. Hence, it is of interest to alter the sampling distribution by designing an alternative inference strategy. Third, we design such an inference strategy–the novel REBASE tree search–and find it is Pareto optimal, in that it achieves the best performance across all tested compute budgets. Notably, it outperforms commonly used weighted majority voting and MCTS methods that have attracted much interest and widespread use. This finding not only shows the strength of REBASE, but also indicates that there is large headroom to improve language model performances via inference-time algorithms.

ACKNOWLEDGMENT

Zhiqing Sun acknowledges the support of the Google Fellowship. Sean Welleck thanks NSF SCALE (NSF DMS 2134012) and Convergent Research.

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

# A OMITTED PROOFS

## A.1 PROOF OF THEOREM 1

*Proof.* Recall that we assume the answer must be shorter than $L$ tokens. Let $\mathcal{A} = \{v \mid |v| \leq L\}$ be the set of all possible answers. Let $\tilde{\pi}(y \mid x)$ be the probability of the language model $\pi$ outputting the answer $y$ to the question $x$ after marginalizing over the "reasoning paths", i.e.,

$$\tilde{\pi}(y \mid x) = \sum_{r \in \mathcal{V}^*} \pi(rey|x).$$

Given an input $x$, Assume that $y^* = \arg\max_{y \in \mathcal{A}} \tilde{\pi}(y|x)$, $y' = \arg\max_{y \in \mathcal{A}\setminus\{y^*\}} \tilde{\pi}(y|x)$, and denote

$$\delta = \tilde{\pi}(y^*|x) - \tilde{\pi}(y'|x).$$

For any $y$, denote by $f_n(y)$ the number of times that the model answers $y$ in the first $n$ samples. Let $E_n$ be the event that majority voting with $n$ samples does not output $y^*$. We note that $E_n$ happens only if there exists $y''$ such that $f_n(y'') \geq f_n(y^*)$. Therefore, by union bound,

$$\begin{aligned}
\mathbb{P}(E_n) &\leq \mathbb{P}(\exists\, y'' \in \mathcal{A}\setminus\{y^*\}, f_n(y'') \geq f_n(y^*)) \\
&\leq \sum_{y'' \in \mathcal{A}\setminus\{y^*\}} \mathbb{P}(f_n(y'') \geq f_n(y^*)) \\
&\leq |\mathcal{A}|\mathbb{P}(f_n(y') \geq f_n(y^*))
\end{aligned}$$

Note that $f_n(y^*) - f_n(y')$ can be viewed as a sum of $n$ i.i.d. random variables, which take value 1 with probability $\tilde{\pi}(y^*|x)$, $-1$ with probability $\tilde{\pi}(y'|x)$, and 0 otherwise. Thus, their expectations are all $\delta = \tilde{\pi}(y^*|x) - \tilde{\pi}(y'|x)$. By Hoeffding's inequality, we have

$$\mathbb{P}(f_n(y') \geq f_n(y^*)) \leq \exp\left(-\frac{n\delta^2}{2}\right).$$

Thus,

$$\mathbb{P}(E_n) \leq |\mathcal{A}| \exp\left(-\frac{n\delta^2}{2}\right) \quad \Rightarrow \quad \sum_{n=1}^{+\infty} \mathbb{P}(E_n) < +\infty.$$

By Borel–Cantelli lemma, we have

$$\mathbb{P}\left(\limsup_{n \to +\infty} E_n\right) = 0,$$

which implies the following is true *almost surely*:

$$\exists\, N \in \mathbb{N}^*, \text{ such that for any } n \geq N, \; y^* = \arg\max_{y \in \mathcal{A}} f_n(y)$$

Hence

$$\lim_{n \to +\infty} \mathrm{acc}_n^{\mathrm{MV}}(\{(x,y)\}; \pi) = \mathbb{I}\,[y = y^*] \qquad \text{(almost surely)}.$$

Recall the definition of $y^*$, the above shows the theorem is true for a dataset with a single example $\{(x,y)\}$. For general datasets $\mathcal{D}$ with $m$ examples, one can apply the above argument to each examples and combine the results to conclude the proof of the almost-sure convergence.

Next, we prove the asymptotic result on $\mathbb{E}\left[\mathrm{acc}_n^{\mathrm{MV}}(\{\mathcal{D}\}; \pi)\right]$. We slightly abuse notation for simplicity as follows: We let $y^*(x_i) = \arg\max_{y \in \mathcal{A}} \tilde{\pi}(y|x_i)$, $y' = \arg\max_{y \in \mathcal{A}\setminus\{y^*(x_i)\}} \tilde{\pi}(y|x_i)$, and let

$$\delta_{\min} = \min_{(x_i, y_i) \in \mathcal{D}} \tilde{\pi}(y^*(x_i)|x_i) - \tilde{\pi}(y'(x_i)|x_i).$$

We denote by $E_n(x_i)$ the event that majority voting with $n$ samples does not output $y^*(x_i)$ given input $x_i$. Then it's easy to see that

$$\mathbb{P}(E_n(x_i)) \leq |\mathcal{A}| \exp\left(-\frac{n\delta_{\min}^2}{2}\right) \quad \Rightarrow \quad \mathbb{P}(E_n(x_i)) = \mathcal{O}(c^{-n})$$

where $c > 1$ is a constant (which does not depend on $i$).

Note that if $\mathrm{acc}_n^{\mathrm{MV}}(\{x_i, y_i\}; \pi) = 1$, we have $y_i = y^*(x_i)$ unless $E_n(x_i)$ happens. In other words,

$$\mathrm{acc}_n^{\mathrm{MV}}(\{x_i, y_i\}; \pi) \leq \mathbb{I}\left[y_i = y^*(x_i)\right] + \mathbb{I}[E_n(x_i)]$$
$$\Rightarrow \quad \left|\mathbb{E}\left[\mathrm{acc}_n^{\mathrm{MV}}(\{x_i, y_i\}; \pi)\right] - \mathbb{I}\left[y_i = y^*(x_i)\right]\right| \leq \mathbb{P}(E_n(x_i)) = \mathcal{O}(c^{-n}).$$

Taking a summation over the entire dataset $\mathcal{D}$ yields

$$\left|\mathrm{acc}_n^{\mathrm{MV}}(\mathcal{D}; \pi) - \frac{1}{m}\sum_{i=1}^m \mathbb{I}\left[y_i = y^*(x_i)\right]\right| \leq \frac{1}{m}\sum_{i=1}^m \mathbb{P}(E_n(x_i)) = \mathcal{O}(c^{-n}),$$

which concludes the proof. □

## A.2 PROOF OF THEOREM 2

*Proof.* The proof is similar to the proof of Theorem 1. We only need to set

$$\tilde{\pi}(y \mid x) = \sum_{r \in \mathcal{V}^*} \pi(r e y | x_i) \rho(x_i r e y).$$

Then the technique in the proof of Theorem 1 immediately applies. □

# B MCTS DETAILS

In this section, we present additional background on the Monte Carlo Tree Search (MCTS) algorithm. The MCTS process can be formulated as the following steps:

**Selection.** The process begins at the root node. Here, the algorithm recursively selects the child node that offers the highest Upper Confidence Bound applied to Trees (UCT) value, continuing until a node is reached that has not been expanded. The UCT is calculated using the formula

$$UCT(s) = Q(s) + C\sqrt{\frac{\ln\left(N(\text{Parent}(s))\right)}{N(s)}},$$

where $Q(s)$ denotes the quality score of node $s$, $N(s)$ is the number of visits to node $s$, $\text{Parent}(s)$ denotes the parent node of $s$, and $C$ is a constant determining the level of exploration.

**Expansion and evaluation.** Upon reaching a non-terminal node $s$, the node is expanded by generating multiple child nodes. Each child node $c$ is then evaluated using a value function $V(c)$, which predicts the potential quality of continuing the sequence from node $c$.

**Backpropagation.** After evaluation, the algorithm updates the UCT values and the visit counts for all nodes along the path from the selected node back to the root. For any node $n$ in this path, the updates are made as follows:

$$N(n) \leftarrow N(n) + 1,$$
$$Q(n) \leftarrow \frac{(N(n) - 1)\,Q(n) + V(s)}{N(n)}.$$

## C  HYPER-PARAMETERS

**Finetuning.**  All the hyperparameters for model fine-tuning can be found in Tab. 2. We preprocess the MetaMath Dataset to make the solutions in a stepwise format.

Table 2: Fine-tuning Hyper-parameters: LR refers to the learning rate, BS refers to the batch size. Pythia, Llemma-7B and LLemma-34B are the generators we use in our experiments, RM is short for Reward Model. We only use problems from GSM8K to train the Pythia models.

| Model | # Epoch | Dataset | BS | LR | Max Seq Length | Dtype |
|---|---|---|---|---|---|---|
| Pythia-410M | 1 | MetaMath (GSM8K) | 128 | 8E-5 | 768 | FP32 |
| Pythia-1.4B | 1 | MetaMath (GSM8K) | 128 | 4E-5 | 768 | FP32 |
| Pythia-2.8B | 1 | MetaMath (GSM8K) | 128 | 3E-5 | 768 | FP32 |
| Pythia-6.9B | 1 | MetaMath (GSM8K) | 128 | 2E-5 | 768 | FP32 |
| Pythia-12B | 1 | MetaMath (GSM8K) | 128 | 1E-5 | 768 | FP32 |
| Llemma-7B | 1 | MetaMath | 128 | 8E-6 | 1024 | FP32 |
| Llemma-34B | 1 | MetaMath | 128 | 8E-6 | 768 | FP32 |
| Llemma-34B RM | 2 | Math-Shepherd | 128 | 1E-5 | 768 | BF16 |

**Inference.**  For all the inference strategies, the temperature for LLM token generation is set to $1.0$. Max tokens for the output is $1024$ and max tokens for one step is $256$. For REBASE, we set the balance temperature (i.e., the parameter $T_b$ in Eq. (1)) to $0.1$. For MCTS, we set $C$ in the UCT value to 1 and we expand $4, 8, 16$ children for the root, 2 children for other selected nodes with total $32, 64, 128$ expansions respectively. New expanded nodes will be assigned values by the PRM, and then backpropagate the $Q$ values through the process described in the last section.

# D ADDITIONAL EXPERIMENTAL RESULTS

## D.1 MAJORITY VOTING EXPERIMENT RESULTS

In this section, we additionally include experimental results on the majority voting method, along with its comparison with weighted majority voting (Fig. 9, 10 ,11, 12). The experiments show that although the gap between majority voting and weighted majority voting on sampling is huge. This gap becomes much smaller if we apply REBASE. This phenomenon can be caused by the selection ability of tree search like REBASE. Once REBASE already samples solutions with high rewards, conducing weighted majority voting gains less since the sampled solutions may all have relatively high and stable rewards compared with those of sampling.

## D.2 ADDITIONAL EXPERIMENTS ON LLAMA3 MODELS

We conduct additional experiments with Llama3-8B-Instruct (Dubey et al., 2024) model on MATH and GSM8K datasets, as shown in Fig. 13. Results for code generation task MBPP are presented (Austin et al., 2021) in Tab. 3. These experiments demonstrate that our conclusions generalize to the Llama3 architecture and coding tasks, confirming that increased computational effort improves performance until saturation is reached, and REBASE reaches the optimal performance-compute trade-off.

In mathematical reasoning tasks, REBASE consistently outperforms the sampling approach across different answer selection strategies, including best-of-$n$, majority voting, and weighted voting. The highest performance on each dataset is achieved using REBASE. Specifically, on GSM8K, REBASE combined with weighted majority voting using 128 samples achieves an accuracy of $90.2\%$, surpassing the best accuracy of $89.7\%$ obtained by the sampling method with 256 samples using the best-of-$n$ strategy. Similarly, on MATH, REBASE with weighted majority voting using 128 samples

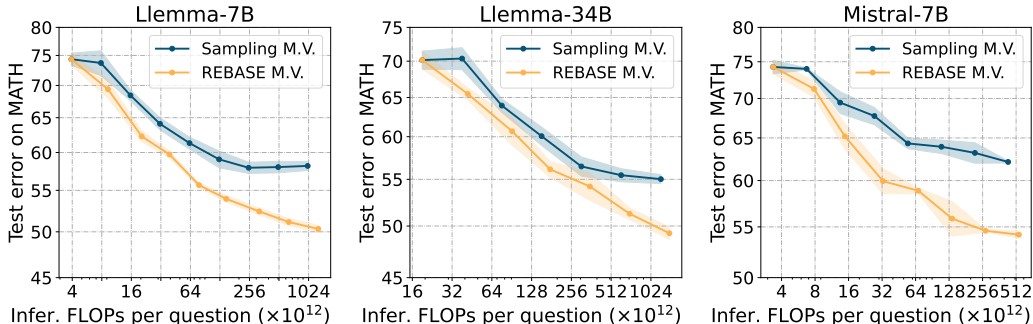

Figure 9: **The inference scaling laws** of different models for the problem-solving error rate on **MATH** test set. The tested models are Llemma-7B (left), Llemma-34B (middle), & Mistral-7B (right). In the legend, M.V. refer to majority voting.

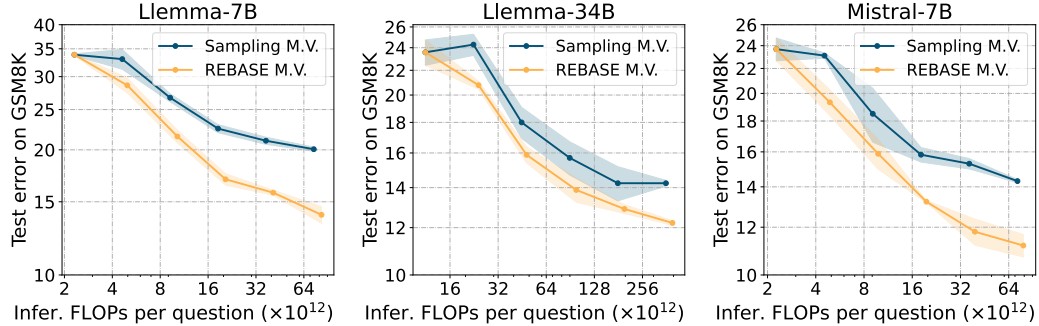

Figure 10: **The inference scaling laws** of different models for the problem-solving error rate on **GSM8K** test set. The tested models are Llemma-7B (left), Llemma-34B (middle), & Mistral-7B (right). In the legend, M.V. refer to majority voting.

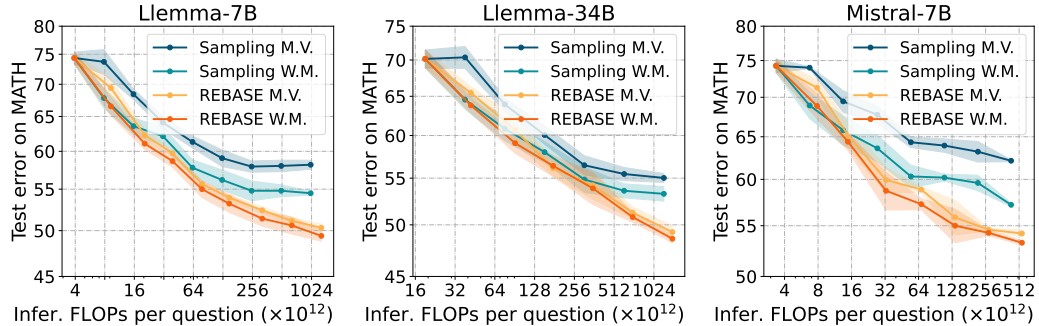

Figure 11: **The inference scaling laws** of different models for the problem-solving error rate on **MATH** test set. The tested models are Llemma-7B (left), Llemma-34B (middle), & Mistral-7B (right). In the legend, M.V. and W.M. refer to majority voting and weighted majority, respectively.

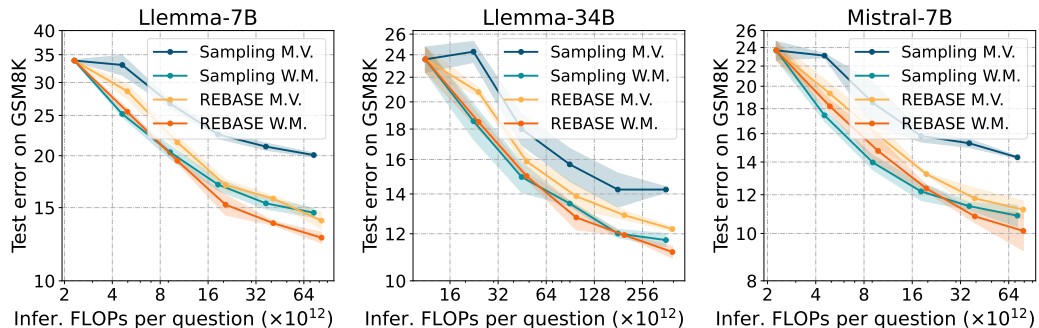

Figure 12: **The inference scaling laws** of different models for the problem-solving error rate on **GSM8K** test set. The tested models are Llemma-7B (left), Llemma-34B (middle), & Mistral-7B (right). In the legend, M.V. and W.M. refer to majority voting and weighted majority, respectively.

Table 3: Zero shot pass rates of Sampling and REBASE on MBPP code generation task.

| # Samples | Sampling FLOPs | Sampling Pass Rate | REBASE FLOPs | REBASE Pass Rate |
|---|---|---|---|---|
| 8 | $8 \times 10^{12}$ | 63 | $8.3 \times 10^{12}$ | 69.6 |
| 16 | $16 \times 10^{12}$ | 69.4 | $17.5 \times 10^{12}$ | 72.4 |
| 32 | $32 \times 10^{12}$ | 72.4 | $34.9 \times 10^{12}$ | 75.8 |
| 64 | $64 \times 10^{12}$ | 79 | $69.2 \times 10^{12}$ | 81.4 |

achieves an accuracy of $47.4\%$, significantly outperforming the sampling method's best accuracy of $41.9\%$ with 256 samples using best-of-$n$.

For the code generation task MBPP, we analyze scaling behavior and compute-optimal inference through pass rate evaluation. The results confirm that REBASE is more compute-efficient than sampling. This advantage can be attributed to the use of a reward model that evaluates partial code solutions. By conducting one iteration of REBASE, our method prunes suboptimal partial solutions while encouraging exploration of promising ones, thereby enhancing computational efficiency and solution quality.

### D.3 COMPARISON OF DIFFERENT STRATEGIES ACROSS DIFFERENT MODELS

We show the accuracy of different strategies under a specific compute budget in Tab. 4.

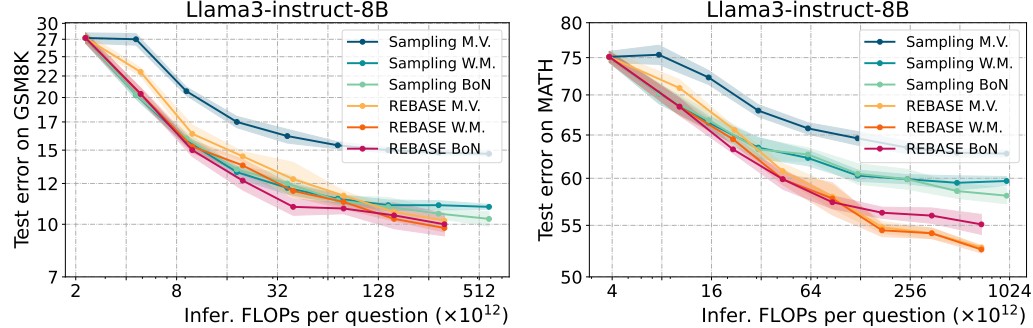

Figure 13: **GSM8K (left) and MATH (right) inference scaling across inference strategies and models** (lower is better). The tested model is Llama3-instruct-8B. In the legend, M.V., W.M., and BoN refer to majority voting, weighted majority, and best-of-$n$, respectively.

Table 4: **Accuracy of different inference configurations under a specific compute budget**. MV, BoN and WV denote Majority Voting, best-of-$n$ and Weighted Voting, respectively.

| | # SAMPLES | MATH FLOPs | GSM8K FLOPs | MATH500 | GSM8K |
|---|---|---|---|---|---|
| **MISTRAL-7B** | | | | | |
| GREEDY | 1 | $3.4 \times 10^{12}$ | $2.3 \times 10^{12}$ | 28.6 | 77.9 |
| SAMPLING + MV | 32 | $109.2 \times 10^{12}$ | $72.6 \times 10^{12}$ | 36.1 | 85.7 |
| SAMPLING + BoN | 32 | $109.2 \times 10^{12}$ | $72.6 \times 10^{12}$ | 40.3 | 89.4 |
| SAMPLING + WV | 32 | $109.2 \times 10^{12}$ | $72.6 \times 10^{12}$ | 39.7 | 89.1 |
| REBASE + MV | 32 | $136.2 \times 10^{12}$ | $78.9 \times 10^{12}$ | 44.1 | 88.8 |
| REBASE + BoN | 32 | $136.2 \times 10^{12}$ | $78.9 \times 10^{12}$ | **45.4** | 89.4 |
| REBASE + WV | 32 | $136.2 \times 10^{12}$ | $78.9 \times 10^{12}$ | 45.0 | **89.8** |
| **LLEMMA-7B** | | | | | |
| GREEDY | 1 | $3.92 \times 10^{12}$ | $2.3 \times 10^{12}$ | 30.0 | 68.5 |
| SAMPLING + MV | 32 | $125.4 \times 10^{12}$ | $73.9 \times 10^{12}$ | 41.0 | 80.0 |
| SAMPLING + BoN | 32 | $125.4 \times 10^{12}$ | $73.9 \times 10^{12}$ | 41.7 | 85.6 |
| SAMPLING + WV | 32 | $125.4 \times 10^{12}$ | $73.9 \times 10^{12}$ | 43.5 | 85.4 |
| REBASE + MV | 32 | $148.0 \times 10^{12}$ | $82.6 \times 10^{12}$ | 46.1 | 86.1 |
| REBASE + BoN | 32 | $148.0 \times 10^{12}$ | $82.6 \times 10^{12}$ | 44.1 | 86.9 |
| REBASE + WV | 32 | $148.0 \times 10^{12}$ | $82.6 \times 10^{12}$ | **46.8** | **87.3** |
| **LLAMA3-INSTRUCT-8B** | | | | | |
| GREEDY | 1 | $3.84 \times 10^{12}$ | $2.28 \times 10^{12}$ | 29.6 | 79.0 |
| SAMPLING + MV | 32 | $122.9 \times 10^{12}$ | $73.2 \times 10^{12}$ | 35.4 | 84.6 |
| SAMPLING + BoN | 32 | $122.9 \times 10^{12}$ | $73.2 \times 10^{12}$ | 39.7 | 88.5 |
| SAMPLING + WV | 32 | $122.9 \times 10^{12}$ | $73.2 \times 10^{12}$ | 39.5 | 88.6 |
| REBASE + MV | 32 | $172.8 \times 10^{12}$ | $79.3 \times 10^{12}$ | 45.2 | 88.3 |
| REBASE + BoN | 32 | $172.8 \times 10^{12}$ | $79.3 \times 10^{12}$ | **45.5** | 88.7 |
| REBASE + WV | 32 | $172.8 \times 10^{12}$ | $79.3 \times 10^{12}$ | 43.7 | **89.1** |
| **LLEMMA-34B** | | | | | |
| GREEDY | 1 | $19.0 \times 10^{12}$ | $11.2 \times 10^{12}$ | 33.0 | 78.4 |
| SAMPLING + MV | 8 | $152.3 \times 10^{12}$ | $89.7 \times 10^{12}$ | 39.9 | 84.3 |
| SAMPLING + BoN | 8 | $152.3 \times 10^{12}$ | $89.7 \times 10^{12}$ | 40.4 | 86.7 |
| SAMPLING + WV | 8 | $152.3 \times 10^{12}$ | $89.7 \times 10^{12}$ | 41.0 | 86.0 |
| REBASE + MV | 8 | $176.8 \times 10^{12}$ | $98.7 \times 10^{12}$ | **43.9** | 86.1 |
| REBASE + BoN | 8 | $176.8 \times 10^{12}$ | $98.7 \times 10^{12}$ | 43.6 | **86.9** |
| REBASE + WV | 8 | $176.8 \times 10^{12}$ | $98.7 \times 10^{12}$ | 42.9 | **86.9** |

