# OpenReview forum: "Inference Scaling Laws: An Empirical Analysis of Compute-Optimal Inference for LLM Problem-Solving"
_ICLR.cc/2025/Conference — ICLR 2025 Poster_

### Official Review · Reviewer_oZkX · 2024-10-21

**Soundness:** 3
**Presentation:** 4
**Contribution:** 4
**Rating:** 8
**Confidence:** 4

**Summary:**

This paper presents an empirical analysis of inference scaling laws for large language models, focusing on the trade-offs between model sizes and generating additional tokens with different inference strategies. The authors explore how varying the compute during inference affects model performance after the model has been trained. They compare various inference techniques, such as greedy search, majority voting, best-of-n, weighted voting, and two tree search algorithms, using different model sizes and compute budgets.

**Strengths:**

S1. The standpoint of this paper is interesting and can provide some insights into the LLM community.

S2. The paper provides an extensive empirical study on the relationship between model size, inference computation, and performance, which is valuable for both research and practical deployment of LLMs.

S3. The paper analyzes not only the performance of different inference strategies but also provides a theoretical analysis of voting methods, offering insights into their convergence behavior.

S4. The experimental results are clearly presented, making it easy to understand the performance trade-offs between different models and inference strategies.

**Weaknesses:**

However, there remain some concerns for me:

W1. The analysis is focused on mathematical reasoning tasks, which may limit the generalizability of the findings to other types of problems. In detail, the paper does not provide extensive discussion on how these findings might generalize to other domains beyond mathematical reasoning.

W2. The study is based on specific two LLMs, and it's unclear how the results would extend to other model architectures.

W3. Some of the figures are a bit unclear. For example, in Figures 2 and 3, although the description is vivid, the set font is a bit hard to read.

**Questions:**

Q1: How well do the findings generalize to non-mathematical reasoning tasks, such as natural language inference or summarization?

Q2: How sensitive are the results to the specific model architectures used in the study? Would different architectures yield similar findings?

---

> ### Author Response · Authors · 2024-11-21
> **Rebuttal**
>
> We thank you for taking the time to review our paper and are very glad to see you like our paper. We appreciate that you find our paper to be insightful and our experiments to be comprehensive. We have added additional experiments on model Llama-3-Instruct and on coding task MBPP to address your concerns. We include the discussions, figures and results in appendix D and will move them to the main part in later revisions.
>
> **Regarding generalization to the tasks beyond mathematical reasoning.** Thank you for your constructive comment. We additionally conduct experiments on the code generation task MBPP using the Llama3-Instruct 8B model. The results are shown below:
>
> **Llama3-Instruct 8B on MBPP**:
> | # of Samples | Sampling FLOPS ($\times 10^{12}$) | Sampling Pass@n | REBASE FLOPS ($\times 10^{12}$) | REBASE Pass@n |
> |-------|-------------|--------------|----------|-----------|
> | 8     | 8   | 63%      | 8.26  | 69.6%   |
> | 16    | 16    | 69.4%     | 17.47      | 72.4%  |
> | 32    | 32   | 72.4%    | 34.9   | 75.8%  |
> | 64  | 64   | 79%    | 69.15   | 81.4%    |
>
> The above results are consistent with our findings on compute-optimal inference: Well-designed inference algorithms like REBASE surpasses widely used sampling methods, indicating the large potential of advanced inference algorithms for better inference scaling on diverse tasks.
>
> **Regarding the sensitivity to model architectures**:
>
> Thank you for the question. Our original experiments have tested three different model families with different architectures, i.e., Pythia, Mistral, and Llemma. We further extend our study to Llama3 and our additional experimental results are shown below:
>
> **Llama3-Instruct 8B MATH**:
> | # of Samples | Sampling FLOPS ($\times 10^{12}$) | Sampling weighted voting | REBASE FLOPS ($\times 10^{12}$) | REBASE weighted voting|
> |--------|--------|-----------|--------------|---------------|
> | 1    | 3.84  | 24.9%  | 3.84  | 24.9%  |
> | 2   | 7.68  |29.8%  | 10.2  | 31.5%     |
> | 4    | 15.3  |33.5%  | 22.1 | 35.5%    |
> | 8  | 30.7  |36.5%  | 44.2  |40.1%   |
> | 16   | 61.4  | 37.7%  | 88.3  | 42.2%  |
> | 32  | 122  | 39.7%  | 177  | 45.5%   |
> | 64   | 245 | 40.1%  | 353  | 45.8% |
> | 128   | 491 | 40.5%  |706  |47.4%  |
> | 256 | 983 | 40.3%  |   |      |
>
>
>
> | # of Samples | Sampling FLOPS ($\times 10^{12}$) | Sampling Best-of-N | REBASE FLOPS ($\times 10^{12}$) | REBASE Best-of-N|
> |------------|--------|---------------|------------|-------|
> | 1   | 3.84    | 24.9%   | 3.84     | 24.9%  |
> | 2  | 7.68    |29.8%   | 10.2   | 31.5%  |
> | 4  | 15.4    |33.2%   | 22.1   | 36.7%   |
> | 8   | 30.7    |36.7%   | 44.2    |40.1%    |
> | 16 | 61.4       | 37.3%   | 88.3   | 42.6%  |
> | 32    | 122   | 39.5%   | 177 | 43.7%  |
> | 64    | 246   | 40.1%  | 353  | 44%   |
> | 128   | 492  | 41.4%  |706  |44.9%  |
> | 256  | 983    | 41.9%   |     |      |
>
> **Llama3-Instruct 8B GSM8K**:
>
>
> | # of Samples | Sampling FLOPS ($\times 10^{12}$) | Sampling weighted voting | REBASE FLOPS ($\times 10^{12}$) | REBASE weighted voting|
> |---------|---------|-----------|------------|------------|
> | 1     | 2.28   |72.3%    | 2.28      | 72.3% |
> | 2     | 4.58     |79.7%     | 4.93   | 79.6%  |
> | 4   | 9.15    |84.2%    | 9.92    | 84.7%  |
> | 8  | 18.3   |86.7%    | 19.8    |86.2%   |
> | 16   |36.6    | 87.8%      |39.7     | 88%   |
> | 32    | 73.2    | 88.5%     | 79.4  | 88.7%   |
> | 64    | 146   | 88.9%    | 159   | 89.7%  |
> | 128   | 293  | 88.9%     |317   |90.2%   |
> | 256  | 586  | 89%   |     |      |
>
> | # of Samples | Sampling FLOPS ($\times 10^{12}$) | Sampling Best-of-N | REBASE FLOPS ($\times 10^{12}$) | REBASE Best-of-N|
> |-------|---------|--------|-----------|-----------|
> | 1   | 2.28 | 72.3% | 2.28| 72.3% |
> | 2   | 4.58 |79.7%   | 4.93  |79.6%    |
> | 4    | 9.15  |84%   | 9.92  | 85%   |
> | 8   | 18.3 |86.5%  | 19.8   |87.3%    |
> | 16    |36.6    | 87.5% |39.7  | 89%   |
> | 32    | 73.2     | 88.6%  | 79.4   | 89.1%  |
> | 64   | 146  | 89% | 159  | 89.5% |
> | 128   | 293   | 89.4%  |317 |90%  |
> | 256 | 586 | 89.7% |         |      |
>
> We find that experimental results on Llama3 share similar trends with our original observation. In particular, REBASE still offers non-trivial performance gains on the Llama3 model, especially on the challenging MATH dataset. We are grateful for your suggestion, which helps us better demonstrate the robustness of our findings and keep the research up-to-date.
>
> **Regarding the presentation of the figures.** Thank you for the feedback. We will update those figures for better clarity.
>
> We sincerely hope that our responses address your concerns and you reevaluate our work based on the responses. Thank you again for your time!

---

> > ### Comment · Reviewer_oZkX · 2024-11-24
> >
> > Thanks for the reply. It has addressed my concerns well. I have raised my score to support this work.
> >
> > Further, it would be excellent to include more LLM models besides Llama-3-Instruct in the future to provide more insights to the research community.

---

> > > ### Author Response · Authors · 2024-11-25
> > > **Thank you for raising your score!**
> > >
> > > We sincerely appreciate your kind support and raised score - it means a great deal to us! We are actively continuing our research on inference scaling and exploring more models and tasks in this domain. We will share more updated results in our paper in future revisions!

---

### Official Review · Reviewer_2jwG · 2024-11-04

**Soundness:** 3
**Presentation:** 4
**Contribution:** 2
**Rating:** 6
**Confidence:** 3

**Summary:**

In this work, the authors study how varying the amount of compute utilized during inference (i.e., varying the number of samples collected during decoding) affects model accuracy on a task. In particular, they explore how to maximize accuracy given a specific compute budget (measured in FLOPs). They experiment with Pythia, Mistral and Llemma models on 2 mathematical reasoning tasks. In experiments, they vary the inference strategy among sampling and tree search methods, coupled with best-of-n selection or weighted majority voting among the generated candidate sets.  Their results show that in many cases, under small compute budgets, smaller models with more compute-intensive inference yields the best results; under large compute budgets, larger models perform better. The results also suggest that their proposed tree-search method–ReBASE–tends to yield the optimal compute-accuracy tradeoff across compute budgets and models. The paper also includes an asymptotic analysis of voting methods as the number of samples goes to infinity. Based on these results, the authors defend a need for more sophisticated inference algorithms.

**Strengths:**

**Topic and Formalization.** I think the question being explored in this paper is interesting and timely–especially considering recent papers, such as those discussed in the concurrent work section. I also find the formalization–maximizing performance under a compute budget–attractive and satisfying. As such, the work is well-motivated, easy to follow and likely to be relevant to many.  Practitioners would be well-suited to consider this type of analysis when deciding what models to deploy.

**Well-designed experiments & complete analysis.** For the experiments that were run, the authors did a good job designing the experiments, and analyzing and visualizing results. The methods selected for comparison are good. I believe that the compute budgets selected are _likely_ good however it would be helpful to report the number of samples drawn for each model/method under each compute budget to make sure they cover a sufficiently large, yet realistic, number of samples.  Generally, it is easy to use the visualizations to see which methods and models are best under various compute budgets. In addition to the discussion of the main results, the discussion of the affect of tree search on weaker models as well as performance saturation adds to the paper. The results make sense and are likely useful. I appreciate the inclusion of error bars.

**Well-written.** The writing is clear and the paper is easy to follow.

Overall, the paper exhibits a number of good characteristics. It’s well written, the topic is interesting, it provides some theoretical analysis, the experiments are solid and the results are relevant.

**Weaknesses:**

**Mathematics Tasks Only.**  The experiments only study two datasets, both of which are related to mathematical reasoning. As such, I’m concerned about the generalizability to new domains and new tasks. I think this paper could be strengthened with experiments on additional tasks like question answering and/or its variants.

**Limited models tested.**  The set of models tested leaves something to be desired. In my understanding and experience, Pythia models tend to be weak in comparison to newer open-source models, such as the Llama3.1 family of models. Lemma is a special purpose model for the mathematical domain; Mistral-7B is generally understood to be a strong model.  As such, I’m a bit worried that the conclusions would not extend to better models. As is reported, the proposed method, ReBASE, tends to help the weakest models and might not provide the same advantages for a model like Llama3.1-7b-instruct.

**Some results seem insignificant.** Figure 7 shows a lot of overlap amongst the methods tested. This is especially the case for Mistral-7B (which is perhaps the strongest model tested).  This overlap exists in Figure 6 as well; albeit, less so. Again, this makes me concerned about how much the results extend to better models. And, in Figures 4 and 5, ReBASE with the larger models doesn’t seem to be significantly better than sampling.

Overall, I’m concerned about the generalizability of the results to new tasks, and the extent to which the results would hold for the most performant models.

**Questions:**

- What is the difference between ReBASE and beam search with sampling turned on? They appear to be quite similar.
- Not a question, but I think that you may want to discuss the “Dot-by-dot” paper (https://arxiv.org/abs/2404.15758) since it also concludes that generating more tokens can be helpful; albeit their setup is much different from yours.
- In the expression denoting the reward of a node r_k (last line of page 6), what was q?
- In Figure 5, what happens if you continue sampling from the 7B model with ReBASE until the maximum number of FLOPs considered (i.e., if we extend the orange line all the way to the right, what does it look like)?
- Why is test error reported instead of task accuracy? Please report the test accuracy for the sake of comparison to other work.
- Can you report the number of samples you were able to draw from each model / inference method corresponding to the number of FLOPs used?

---

> ### Author Response · Authors · 2024-11-21
> **Rebuttal part (1/3)**
>
> We thank you for the time for reviewing and the constructive feedback! We are glad to see you like our paper and we appreciate that you have found the topics in our paper attractive and important. We address your concerns below.
>
> > Mathematics Tasks Only. The experiments only study two datasets, both of which are related to mathematical reasoning. As such, I’m concerned about the generalizability to new domains and new tasks. I think this paper could be strengthened with experiments on additional tasks like question answering and/or its variants.
>
> We add results on a code task, MBPP, to show that REBASE is also compute optimal compared to sampling on the code benchmark MBPP.
>
> **Llama3-Instruct 8B on MBPP**:
> | # of Samples | Sampling FLOPS ($\times 10^{12}$) | Sampling Pass@n | REBASE FLOPS ($\times 10^{12}$) | REBASE Pass@n |
> |-------|-------------|------------|----------|-----------|
> | 8     | 8   | 63%      | 8.26  | 69.6%   |
> | 16    | 16    | 69.4%     | 17.47      | 72.4%  |
> | 32    | 32   | 72.4%    | 34.9   | 75.8%  |
> | 64  | 64   | 79%    | 69.15   | 81.4%    |
>
>
>
> > Limited models tested. The set of models tested leaves something to be desired. In my understanding and experience, Pythia models tend to be weak in comparison to newer open-source models, such as the Llama3.1 family of models. Lemma is a special purpose model for the mathematical domain; Mistral-7B is generally understood to be a strong model. As such, I’m a bit worried that the conclusions would not extend to better models. As is reported, the proposed method, ReBASE, tends to help the weakest models and might not provide the same advantages for a model like Llama3.1-7b-instruct.
>
> We additionally conduct experiments using Llama-3-8B-Instruct, and report the accuracy and update the figures in the paper. The results are averaged over multiple independent experiments. We also add the figures and discussions in the appendix D and will move them to the main part in later revisions.
>
> **Llama3-Instruct 8B MATH**:
> | # of Samples | Sampling FLOPS ($\times 10^{12}$) | Sampling weighted voting | REBASE FLOPS ($\times 10^{12}$) | REBASE weighted voting|
> |--------|--------|-----------|--------------|---------------|
> | 1    | 3.84  | 24.9%  | 3.84  | 24.9%  |
> | 2   | 7.68  |29.8%  | 10.2  | 31.5%     |
> | 4    | 15.3  |33.5%  | 22.1 | 35.5%    |
> | 8  | 30.7  |36.5%  | 44.2  |40.1%   |
> | 16   | 61.4  | 37.7%  | 88.3  | 42.2%  |
> | 32  | 122  | 39.7%  | 177  | 45.5%   |
> | 64   | 245 | 40.1%  | 353  | 45.8% |
> | 128   | 491 | 40.5%  |706  |47.4%  |
> | 256 | 983 | 40.3%  |   |      |
>
>
>
> | # of Samples | Sampling FLOPS ($\times 10^{12}$) | Sampling Best-of-N | REBASE FLOPS ($\times 10^{12}$) | REBASE Best-of-N|
> |------------|--------|---------------|------------|-------|
> | 1   | 3.84    | 24.9%   | 3.84     | 24.9%  |
> | 2  | 7.68    |29.8%   | 10.2   | 31.5%  |
> | 4  | 15.4    |33.2%   | 22.1   | 36.7%   |
> | 8   | 30.7    |36.7%   | 44.2    |40.1%    |
> | 16 | 61.4       | 37.3%   | 88.3   | 42.6%  |
> | 32    | 122   | 39.5%   | 177 | 43.7%  |
> | 64    | 246   | 40.1%  | 353  | 44%   |
> | 128   | 492  | 41.4%  |706  |44.9%  |
> | 256  | 983    | 41.9%   |     |      |
>
> **Llama3-Instruct 8B GSM8K**:
>
>
> | # of Samples | Sampling FLOPS ($\times 10^{12}$) | Sampling weighted voting | REBASE FLOPS ($\times 10^{12}$) | REBASE weighted voting|
> |---------|---------|-----------|------------|------------|
> | 1     | 2.28   |72.3%    | 2.28      | 72.3% |
> | 2     | 4.58     |79.7%     | 4.93   | 79.6%  |
> | 4   | 9.15    |84.2%    | 9.92    | 84.7%  |
> | 8  | 18.3   |86.7%    | 19.8    |86.2%   |
> | 16   |36.6    | 87.8%      |39.7     | 88%   |
> | 32    | 73.2    | 88.5%     | 79.4  | 88.7%   |
> | 64    | 146   | 88.9%    | 159   | 89.7%  |
> | 128   | 293  | 88.9%     |317   |90.2%   |
> | 256  | 586  | 89%   |     |      |
>
> | # of Samples | Sampling FLOPS ($\times 10^{12}$) | Sampling Best-of-N | REBASE FLOPS ($\times 10^{12}$) | REBASE Best-of-N|
> |-------|---------|--------|-----------|-----------|
> | 1   | 2.28 | 72.3% | 2.28| 72.3% |
> | 2   | 4.58 |79.7%   | 4.93  |79.6%    |
> | 4    | 9.15  |84%   | 9.92  | 85%   |
> | 8   | 18.3 |86.5%  | 19.8   |87.3%    |
> | 16    |36.6    | 87.5% |39.7  | 89%   |
> | 32    | 73.2     | 88.6%  | 79.4   | 89.1%  |
> | 64   | 146  | 89% | 159  | 89.5% |
> | 128   | 293   | 89.4%  |317 |90%  |
> | 256 | 586 | 89.7% |         |      |
>
> We find that experimental results on Llama3 share similar trends with our original observation. In particular, REBASE still offers non-trivial performance gains on the Llama3 model, especially on the challenging MATH dataset. We are grateful for your suggestion, which helps us better demonstrate the robustness of our findings and keep the research up-to-date.

---

> ### Author Response · Authors · 2024-11-21
> **Rebuttal part (2/3)**
>
> > Some results seem insignificant. Figure 7 shows a lot of overlap amongst the methods tested. This is especially the case for Mistral-7B (which is perhaps the strongest model tested). This overlap exists in Figure 6 as well; albeit, less so. Again, this makes me concerned about how much the results extend to better models. And, in Figures 4 and 5, ReBASE with the larger models doesn’t seem to be significantly better than sampling.
>
> Overlaps typically happen in the leftmost parts of the figures. In each scaling curve, the result begins from only one sample. Thus, these leftmost parts indeed correspond to comparing different methods using only 1-4 sample(s). In such cases, tree search approaches almost degenerate into naive sampling, and overlapped curves are expected.
>
> We agree with you that REBASE offers larger performance gains on MATH (Fig. 6) than GSM8K (Fig. 7). We believe that REBASE can be more advantageous on more challenging datasets. To elaborate, the average length of the model-generated answers on GSM8K is $2\times$ shorter than that on MATH. This indicates that solving problems on MATH requires longer reasoning chains, i.e., more steps, compared with GSM8K. Our results show that REBASE is more advantageous under scenarios where more reasoning steps are required, demonstrating the effectiveness of the search strategy. This finding also suggests that the notion of compute-optimal inference can depend on the data property.
>
> In Figs. 4 & 5, we believe that REBASE consistently shows decent performance gains. For example, our proposed REBASE leads to 5.3%, 3.3%, and 2.6% performance gains on MATH for Mistral-7B, Llemma-7B, Llemma-34B, respectively. Our additional experiments on Llama3 above also exhibit the same trend, demonstrating the robustness of the finding.
>
>
> > What is the difference between ReBASE and beam search with sampling turned on? They appear to be quite similar.
>
> Beam search expands to an equal number of next nodes given current partial solutions and retains a fixed number of highest solutions. Each partial solution is explored equally. In contrast, REBASE is encouraged to explore the partial solutions with higher rewards and prune less promising ones.
>
> Besides, to generate $N$ samples with beam search, in each iteration, $NK$ nodes will be expanded where $K$ denotes the beam size. In contrast, REBASE will only explore at most $N$ nodes, keeping the tree width constant by careful pruning.
>
> Therefore, REBASE can be more computationally efficient and more accurate compared with beam search.
>
> > Not a question, but I think that you may want to discuss the “Dot-by-dot” paper (https://arxiv.org/abs/2404.15758) since it also concludes that generating more tokens can be helpful; albeit their setup is much different from yours.
>
> Thank you for the reference! It is indeed a nice paper which studies an interesting approach of scaling up inference compute and makes nice connections to complexity theory. We believe the “dot-by-dot” method advances scientific understanding of language model inference. However, from the angle of compute-optimal inference, it may not be the most favorable approach, because standard chain-of-thought reasoning is as powerful as, if not more powerful than, the “dot-by-dot” method. We have added this paper in our references.
>
> > In the expression denoting the reward of a node r_k (last line of page 6), what was q?
>
> Thank you for pointing out the typo. Here, $q$ should have $x$ (the input question) defined in line 320. We have already fixed this in our updated manuscript.
>
> > In Figure 5, what happens if you continue sampling from the 7B model with ReBASE until the maximum number of FLOPs considered (i.e., if we extend the orange line all the way to the right, what does it look like)?
>
> This is a good question. We have additionally tested the 7B model in this setting and updated the figures in our manuscript. We find that the model accuracy can continually improve with more FLOPs. However, the best configuration in this setting is the 34B model with the REBASE algorithm.

---

> ### Author Response · Authors · 2024-11-21
> **Rebuttal part (3/3)**
>
> > Why is test error reported instead of task accuracy? Please report the test accuracy for the sake of comparison to other work.
>
> We report test error following the GPT4 technical report [1] and existing works on training scaling laws [2,3], showing the error (inference) / loss (training) decreases with more computation. We agree that reporting test accuracy can facilitate comparisons to other works. For this sake, we add a table reporting the test accuracy in the appendix in our updated manuscript. We hope this addresses your concern.
>
> [1] Achiam, Josh, et al. "Gpt-4 technical report." arXiv preprint arXiv:2303.08774 (2023).
>
> [2]Hoffmann, Jordan, et al. "Training compute-optimal large language models." arXiv preprint arXiv:2203.15556 (2022).
>
> [3]Kaplan, Jared, et al. "Scaling laws for neural language models." arXiv preprint arXiv:2001.08361 (2020).
>
> > Can you report the number of samples you were able to draw from each model / inference method corresponding to the number of FLOPs used?
>
> In all the figures, the datapoint represents $2^i$ samples, where integer i starts from 0. For 7b models, our results cover from 1 sample to 128 or 256 samples. For the 34B model we have covered results of 1 to 64 samples. We have also added the clarification in the revision.
>
> We sincerely hope that our responses address your concerns and you reevaluate our work based on the responses. Thank you again for your time!

---

> ### Author Response · Authors · 2024-11-25
>
> Dear Reviewer,
>
> Thank you again for your time and effort in reviewing our paper. If you could spare a moment to review our response and share any further comments, it would be greatly appreciated!
>
> Warm regards,
>
> Authors of Paper 6864

---

> ### Comment · Reviewer_2jwG · 2024-11-25
> **Thank you for your responses**
>
> Thank you for your thorough responses and providing additional clarity on the various points/questions I raised and making corresponding updates to your manuscript. I sincerely appreciate your running of additional experiments!
>
> As you mention, in your updated experimentations, the results seem most significant for MATH (and MBPP) and less so for GSM8k.  The results seem to hint at length/complexity of the task affecting the magnitude of the improvements you're seeing. I believe these results will help to strengthen your paper and alleviate some doubts related to stronger models.

---

> > ### Author Response · Authors · 2024-11-26
> >
> > Thank you for your thoughtful reply. We are glad to see that our responses address your questions!
> >
> > To summarize, our work investigates inference scaling law: scaling up inference compute influences mathematical task performance for various model families and sizes. Our findings also demonstrate that small models with strong inference strategies can be compute-optimal compared to naively scaling up model sizes.
> >
> > Our additional experiments further extend the finding to coding tasks and more models. We agree with you that according to our results, compute-optimal inference is more critical for challenging tasks (e.g., on MATH). Indeed, we view this trend in a positive way and believe this insight can be useful for pushing the frontier of LLM capability on extremely hard problems.
> >
> > We hope the revised version better meets your expectations. We would greatly appreciate it if you could consider rasing your score. If you have any further questions or suggestions, please feel free to reach out. Thank you again for your time!

---

### Official Review · Reviewer_EzJx · 2024-11-04

**Soundness:** 2
**Presentation:** 3
**Contribution:** 2
**Rating:** 3
**Confidence:** 5

**Summary:**

The study investigates the inference scaling laws for large language models (LLMs), focusing on finding compute-optimal inference configurations. The authors examine the trade-offs between different model sizes and inference strategies, analyzing how these factors impact computing costs and model performance. Through empirical analysis, the study finds that smaller LLMs can outperform larger models given the same computational resources by employing advanced inference strategies. The paper also introduces a compute-optimal inference algorithm which can help small LLMs achieving better cost-performance tradeoffs.

**Strengths:**

1. The study’s motivation of studying the selection of model size and inference strategy under limited computation budgets is meaningful and justified.
2. The formal definitions and proofs presented in the paper are clear and maintain good readability.
3. The integration of a reward model to guide LLMs' inference process is reasonable.
4. The experimental results are clearly presented and easy to read.

**Weaknesses:**

1. The paper lacks technical contribution. The proposed REBASE framework merely integrates process reward models (PRM) into the existing Monte Carlo Tree Search (MCTS) framework. Considering these components are well-established and the simplicity of the REBASE framework, there are concerns regarding the novelty of the work.
2. While the paper employs a PRM as the reward model within REBASE, it fails to detail the training and inference processes of the reward model. Moreover, it does not address how to ensure the reliability of the rewards generated by this model. An unstable or suboptimal reward model could significantly impact the overall framework, and the authors should consider this aspect.
3. The paper provides theoretical analyses for four inference strategies: greedy search, majority voting, best-of-n, and weighted voting, but does not offer comparative experimental results. A detailed experimental comparison involving different model sizes and inference strategies should be included.
4. Similar to traditional MCTS methods, REBASE necessitates extensive intermediate interactions with LLMs before obtaining the final generated results. Additionally, it introduces the extra PRM component. This could lead to significant resource consumption. The authors should provide data analysis on resource usage and runtime to demonstrate whether REBASE is more efficient than conventional MCTS methods.
5. According to the experiments in Section 4.2, both small and large LLMs show high error rates under limited computation scenarios, suggesting that both models are not good enough in these situations. Claiming that smaller models perform better in such conditions is not sufficiently convincing.
6. The experiments in Section 4.3 show inconsistent performance trends of REBASE compared with sampling methods on the MATH and GSM8K datasets. The authors attribute this inconsistency to differences in dataset difficulty, which seems unconvincing. The authors should provide more credible explanations.
7. In the scenarios where REBASE is purportedly advantageous (i.e., less compute-intensive situations), the performance on the MATH dataset with Llemma-34B and Mistral-7B models is either weaker than or merely comparable to the Sampling method. This indicates that REBASE does not consistently achieve improvements. An explanation for this should be provided.
8. The paper only uses two baselines: sampling and MCTS, without elaborating on their implementation details. It would be beneficial to include the latest baselines and provide detailed introductions and comparisons of the baselines and the proposed REBASE method.

**Questions:**

Please answer the questions in the weakness section.

---

> ### Author Response · Authors · 2024-11-21
> **Rebuttal part 1**
>
> Thank you for taking the time to review our paper. We are glad to see that you find our work studying the different inference strategies and model sizes with various compute budgets is meaningful. We answer each of your questions below.
>
> > The paper lacks technical contribution. The proposed REBASE framework merely integrates process reward models (PRM) into the existing Monte Carlo Tree Search (MCTS) framework. Considering these components are well-established and the simplicity of the REBASE framework, there are concerns regarding the novelty of the work.
>
> We feel that there are some misunderstandings on the scope and contributions of our paper. Besides the REBASE algorithm, we would like to emphasize that our main contributions also include rigorous formulation, theoretical characterization, and empirical evaluation of **inference scaling laws** and **compute-optimal inference**, which have not been formally studied in any prior works and contributed to the main novel technical contributions. To reiterate, our novel insight leads to the following observations:
>
> - Smaller models can outperform larger ones under the same compute budget with more samples, and the compute-optimal inference exhibits a scaling law in model size.
> - Sampling and voting methods have performance limits and diminishing returns (which we justified theoretically in Sec. 3.1 and empirically in Sec. 4).
> - Well-designed inference algorithms like REBASE further supasses widely-used sampling and
> MCTS methods, indicating the large potential of equipping smaller models with advanced inference algorithms for better inference scaling.
>
> We also clarify that REBASE is not an integration of process reward models (PRM) and Monte Carlo Tree Search (MCTS) - it is indeed drastically different from MCTS and significantly better than MCTS given compute budgets.
>
> Finally, we point out that the goal of designing REBASE is not presenting a very complicated tree search method that surpasses any other algorithm. Rather, **it is positioned as a part of investigation on compute-optimal inference, demonstrating that applying well-designed inference strategies is more favorable than scaling up the model size.**

---

> ### Author Response · Authors · 2024-11-21
> **Rebuttal part 2**
>
> > While the paper employs a PRM as the reward model within REBASE, it fails to detail the training and inference processes of the reward model. Moreover, it does not address how to ensure the reliability of the rewards generated by this model. An unstable or suboptimal reward model could significantly impact the overall framework, and the authors should consider this aspect.
>
> Thank you for the comment. We report the training hyper-parameters in Appendix C. More specifically, we finetune Llemma 34B on the Math-shepherd to obtain the PRM [1]. Math-shepherd is a popular dataset with step-wise labels on MATH and GSM8K train split. Following [1], the PRM is trained by predicting the step-wise reward through the logits of the step tag tokens.
>
> Our PRM is trained in a standard way following prior works. While we agree that the quality of reward models can be critical, our work does not aim at optimizing reward model training methods since it is orthogonal to our focus on inference scaling laws and compute-optimal inference. Our results show that building upon existing datasets and reward model training techniques already demonstrates better inference scaling behavior than naively scaling up model sizes, which validates our main claim.
>
> [1] Wang, Peiyi, et al. "Math-shepherd: Verify and reinforce llms step-by-step without human annotations." Proceedings of the 62nd Annual Meeting of the Association for Computational Linguistics (Volume 1: Long Papers). 2024.
>
> > The paper provides theoretical analyses for four inference strategies: greedy search, majority voting, best-of-n, and weighted voting, but does not offer comparative experimental results. A detailed experimental comparison involving different model sizes and inference strategies should be included.
>
> All the figures in our experiment sections are comparative studies from various angles. We note that there are multiple varying factors in our study: FLOPs, model sizes, and inference strategies, resulting in hundreds of possible configurations in total. To conduct meaningful comparisons, we present controlled experiments, presenting both comparisons among different inference strategies with fixed model sizes and comparisons among different model sizes with fixed inference strategies.
>
> In our paper revision, we follow your suggestion and add a table in the appendix, including the results with varying model sizes, inference strategies on both GSM8K and MATH datasets. We hope this addresses your concern.
>
>
>
> > Similar to traditional MCTS methods, REBASE necessitates extensive intermediate interactions with LLMs before obtaining the final generated results. Additionally, it introduces the extra PRM component. This could lead to significant resource consumption. The authors should provide data analysis on resource usage and runtime to demonstrate whether REBASE is more efficient than conventional MCTS methods.
>
> In all our presented results, we evaluate the models on the same devices (A6000 GPUs with 48GB memory), and analyze the model test error given FLOPs consumption. Therefore, all the comparisons have taken both model quality and resource usage into account. For example, Fig. 4 clearly demonstrates REBASE shows better cost-performance trade-off than MCTS. We also point out that reward generation only requires a forward pass on the reward model, which is much more efficient than autoregressive generation and does not contribute to the computational bottleneck.
>
> > According to the experiments in Section 4.2, both small and large LLMs show high error rates under limited computation scenarios, suggesting that both models are not good enough in these situations. Claiming that smaller models perform better in such conditions is not sufficiently convincing.
>
> Our claim is that within a fixed compute budget, using a small LLM can potentially outperform the Larger ones since it can generate more samples. We believe our experiments justify this claim. First, in the experiment, the overall trend is consistent: Smaller models outperform larger ones under the same compute-budget when the budget is relatively limited. Second, our experiments in Section 4.2 demonstrate an accuracy of at least 25% on MATH and 68% on GSM8K, with peak performance reaching approximately 50% on MATH and 90% on GSM8K—results that are non-trivial achievements for language models with simple methods and limited compute. Therefore we believe our comparisons are meaningful and the results are well justified.

---

> ### Author Response · Authors · 2024-11-21
> **Rebuttal part 3**
>
> > The experiments in Section 4.3 show inconsistent performance trends of REBASE compared with sampling methods on the MATH and GSM8K datasets. The authors attribute this inconsistency to differences in dataset difficulty, which seems unconvincing. The authors should provide more credible explanations.
>
> We find that the average length of the model-generated answers on GSM8K is $2\times$ shorter than that on MATH. This indicates that solving problems on MATH requires longer reasoning chains, i.e., more steps, compared with GSM8K. Our results show that REBASE is more advantageous under scenarios where more reasoning steps are required, demonstrating the effectiveness of the search strategy.
>
> To further justify this claim, we additionally conduct evaluations on the hard problems (levels 3-5) in MATH based on weighted majority voting. The results are shown below:
>
> | Setting               |  MATH (hard) Sampling Accuracy | MATH (hard) REBASE Accuracy |
> | --------------------- |  ---------------- | ----------------------- |
> | Llemma 7B 32 samples  | 34.1%     | 37.3%                          |
> | Llemma 7B 64 samples  | 34.3%     | 39.8%                          |
> | Llemma 34B 32 samples | 36.8%    | 40.1%                          |
> | Llemma 34B 64 samples | 36.8%                            | 43.3%                          |
>
>
> The results show that REBASE has stronger performance particularly on hard problems. For example, the sampling accuracy of Llemma 34B does not improve when increasing the sample size from 32 to 64, while REBASE shows better scaling trends. We hope the additional results make our claim more convincing.
>
>
> > In the scenarios where REBASE is purportedly advantageous (i.e., less compute-intensive situations), the performance on the MATH dataset with Llemma-34B and Mistral-7B models is either weaker than or merely comparable to the Sampling method. This indicates that REBASE does not consistently achieve improvements. An explanation for this should be provided.
>
> We believe that you are referring to the leftmost parts of the figures. In each scaling curve, the result begins from only one sample. Thus, these leftmost parts indeed correspond to comparing different methods using only 1-4 sample(s). In such cases, tree search approaches almost degenerate into naive sampling. The FLOPs budget is so limited that there is little room to compare different algorithms.
>
> More meaningful comparisons can be observed in the middle ranges in these figures, where the budget is still limited but offers decent room for applying advanced algorithms. In this regime, we see that REBASE performs consistently better than sampling, e.g., in Fig. 6.
>
> > The paper only uses two baselines: sampling and MCTS, without elaborating on their implementation details. It would be beneficial to include the latest baselines and provide detailed introductions and comparisons of the baselines and the proposed REBASE method.
>
> Thank you for the comment. As has been made clear in the keywords and abstract, the primary focus of our paper is inference scaling laws and compute-optimal inference, rather than designing the world’s best mathematical reasoning algorithms. As the initial step in this direction, we begin with systematic theoretical and empirical evaluations of simple methods. We are also aware of many recent works on new methods for mathematical problem solving with LLMs. Studying them in the context of inference scaling laws and compute-optimal inference can be interesting directions for future research.
>
> For the baselines in our work, we have documented them in Sec. 3 and the Appendix. We plan to release our full code upon paper acceptance. We are also happy to include more details in the Appendix if the reviewer finds it helpful.
>
> We sincerely hope that our responses address your concerns and you reevaluate our work based on the responses. Thank you again for your time!

---

> > ### Author Response · Authors · 2024-11-29
> > **Looking forward to your valuable feedback!**
> >
> > Dear Reviewer EzJx,
> >
> > Thank you again for your time and efforts in reviewing our paper. We have carefully responded to each of your questions. Given that the author-reviewer discussion deadline is approaching, we would greatly appreciate it if you could kindly review our responses and share your valuable feedback. If you have any remaining concerns, we would be more than happy to discuss them further.
> >
> > Best regards,
> >
> > Authors of Paper 6864

---

> ### Author Response · Authors · 2024-11-25
>
> Dear Reviewer,
>
> Thank you again for your time and effort in reviewing our paper. If you could spare a moment to review our response and share any further comments, it would be greatly appreciated!
>
> Warm regards,
>
> Authors of Paper 6864

---

### Official Review · Reviewer_8MgV · 2024-11-05

**Soundness:** 2
**Presentation:** 3
**Contribution:** 3
**Rating:** 6
**Confidence:** 4

**Summary:**

This paper focuses on the intriguing topic inference scaling laws. The authors conducted a detailed analysis using language models of various sizes on math-related datasets, GSM8K and MATH. They discovered that, under the same computation cost, smaller models might outperform larger ones. However, once the performance of smaller models saturates, larger models may achieve better results. These phenomenon is quite significant. Subsequently, the authors carried out theoretical analyses and developed a novel tree search algorithm called REBASE, aiming to achieve a Pareto-optimal trade-off between performance and computational efficiency, following by experiments to demonstrate the effectiveness of the REBASE method.

**Strengths:**

- This paper conducted experimental analysis using language models of varying parameter sizes on math-related datasets GSM8K and MATH. It was observed that smaller models can sometimes outperform larger models in terms of performance, given the same inference computational cost. Furthermore, the authors noted that once the performance of smaller models reaches saturation, their performance tends to be inferior to that of larger models. These observations are significant.
- The author conducted a theoretical analysis, demonstrating that the performance of both standard and weighted majority voting inference strategies will converge when there is an infinite number of samples and the performance after convergence depends solely on the distributions modeled by the language model and the reward model. These theoretical analysis aligns with the experimental results.
- The author proposes a new inference strategy that can achieve comparable or even better performance with less computational overhead than MCTS.
- The organization of the article is clear, and the writing is excellent. It's easy to understand when reading.

**Weaknesses:**

- The authors conducted an analysis of inference scaling laws on Pythia and Llemma. However, they did not perform REBASE experiments on Pythia. Instead, they used Llemma and Mistral-7B for these experiments. It would be more convincing if the author could also conduct REBASE experiments on Pythia.
- This paper only conducts experiments on mathematical datasets. Although the authors acknowledge this limitation, their title is "LLM Problem-Solving" rather than "LLM Math Problem-Solving". In fact, the analysis and method proposed by the authors are not confined to math-related problem-solving. A clear indication of this is that in Section 3 (theoretical analysis and the introduction of the REBASE method), the term "Math" is not mentioned. This suggests that similar phenomena may occur in other problem-solving scenarios. Conducting experiments in a broader range of problem-solving areas, such as coding and logical reasoning or others, would enhance the generalizability of the authors' findings and contribute more significantly to the community.
- The models Pythia, Mistral 7b, and Llemma were introduced quite some time ago. In the past year, more powerful open-source LLMs like Llama3 have been proposed, achieving remarkable advancements across various metrics. It would be beneficial if the author could continue to analyze these newly introduced and widely used models, such as Llama3.

**Questions:**

- The authors analyzed inference scaling laws using Pythia and Llemma but did not include REBASE experiments for Pythia, opting instead for Llemma and Mistral-7B. Including REBASE experiments on Pythia could strengthen the findings.
- The paper focuses on mathematical datasets, yet the title "LLM Problem-Solving" suggests a broader scope beyond math. The methods and analysis in Section 3 do not specifically mention "Math", indicating potential applicability to other areas. Exploring additional problem-solving domains, like coding or logical reasoning, could enhance the study's relevance and impact.
- While the models Pythia, Mistral-7B, and Llemma have been around for some time, recent advancements with models like Llama3 show significant progress. Analyzing these newer models could provide valuable insights and keep the research up-to-date.

---

> ### Author Response · Authors · 2024-11-21
> **Rebuttal**
>
> Thank you for the time for reviewing, we are glad to see you like our paper and find our observations on the inference scaling laws significant. We conduct further experiments to address your concerns and answer your questions below.
>
> > The authors analyzed inference scaling laws using Pythia and Llemma but did not include REBASE experiments for Pythia, opting instead for Llemma and Mistral-7B. Including REBASE experiments on Pythia could strengthen the findings.
>
> We use Pythia models to demonstrate the inference scaling law: Smaller models could outperform larger ones by generating more samples since Pythia models include many different sizes and are relatively small, which could be analyzed given our compute resources. For studies on compute-optimal inference, we used Llemma and Mistral models, which are stronger and more suitable for tree search since they perform well on both MATH and GSM8K. In contrast, Pythia is too weak for the MATH dataset. We also added the Llama-3 8B Instruct model to strengthen our conclusions.
>
> > The paper focuses on mathematical datasets, yet the title "LLM Problem-Solving" suggests a broader scope beyond math. The methods and analysis in Section 3 do not specifically mention "Math", indicating potential applicability to other areas. Exploring additional problem-solving domains, like coding or logical reasoning, could enhance the study's relevance and impact.
>
> Thank you for your valuable suggestion. We follow your suggestion and additionally experiment on a code generation task, MBPP, with Llama3-Instruct-8B models. The results show the potential of using compute-optimal inference methods in other domains, e.g., coding.
>
> **Llama3-Instruct 8B on MBPP**:
> | # of Samples | Sampling FLOPS ($\times 10^{12}$) | Sampling Pass@n | REBASE FLOPS ($\times 10^{12}$) | REBASE Pass@n |
> |---|----|---|-----|-------|
> | 8  | 8  | 63% | 8.26  | 69.6%   |
> | 16  | 16    | 69.4% | 17.47| 72.4%  |
> | 32  | 32   | 72.4%  | 34.9  | 75.8%  |
> | 64  | 64   | 79% | 69.15 | 81.4% |
>
> > While the models Pythia, Mistral-7B, and Llemma have been around for some time, recent advancements with models like Llama3 show significant progress. Analyzing these newer models could provide valuable insights and keep the research up-to-date.
>
> Thank you for your comment. We have added the Llama3-Instruct 8B results in the paper appendix D and also show the results here for your reference.
>
> **Llama3-Instruct 8B MATH**:
> | # of Samples | Sampling FLOPS ($\times 10^{12}$) | Sampling weighted voting | REBASE FLOPS ($\times 10^{12}$) | REBASE weighted voting|
> |--|--|--|---|---|
> | 1    | 3.84  | 24.9%  | 3.84  | 24.9%  |
> | 2   | 7.68  |29.8%  | 10.2  | 31.5%     |
> | 4    | 15.3  |33.5%  | 22.1 | 35.5%    |
> | 8  | 30.7  |36.5%  | 44.2  |40.1%   |
> | 16   | 61.4  | 37.7%  | 88.3  | 42.2%  |
> | 32  | 122  | 39.7%  | 177  | 45.5%   |
> | 64   | 245 | 40.1%  | 353  | 45.8% |
> | 128   | 491 | 40.5%  |706  |47.4%  |
> | 256 | 983 | 40.3%  |   |      |
>
>
>
> | # of Samples | Sampling FLOPS ($\times 10^{12}$) | Sampling Best-of-N | REBASE FLOPS ($\times 10^{12}$) | REBASE Best-of-N|
> |---|--|----|---|-------|
> | 1   | 3.84  | 24.9%   | 3.84     | 24.9%  |
> | 2  | 7.68  |29.8%   | 10.2   | 31.5%  |
> | 4  | 15.4 |33.2%   | 22.1   | 36.7%   |
> | 8   | 30.7 |36.7%   | 44.2    |40.1%    |
> | 16 | 61.4 | 37.3%   | 88.3   | 42.6%  |
> | 32    | 122  | 39.5%   | 177 | 43.7%  |
> | 64    | 246 | 40.1%  | 353  | 44%   |
> | 128   | 492 | 41.4%  |706  |44.9%  |
> | 256  | 983    | 41.9%   |   |  |
>
> **Llama3-Instruct 8B GSM8K**:
>
> | # of Samples | Sampling FLOPS ($\times 10^{12}$) | Sampling weighted voting | REBASE FLOPS ($\times 10^{12}$) | REBASE weighted voting|
> |-----|--|---|---|------------|
> | 1   | 2.28   |72.3%    | 2.28      | 72.3% |
> | 2   | 4.58     |79.7%     | 4.93   | 79.6%  |
> | 4   | 9.15    |84.2%    | 9.92    | 84.7%  |
> | 8  | 18.3   |86.7%    | 19.8    |86.2%   |
> | 16   |36.6    | 87.8%      |39.7     | 88%   |
> | 32    | 73.2    | 88.5%     | 79.4  | 88.7%   |
> | 64    | 146   | 88.9%    | 159   | 89.7%  |
> | 128   | 293  | 88.9%     |317   |90.2%   |
> | 256  | 586  | 89%   |     |      |
>
> | # of Samples | Sampling FLOPS ($\times 10^{12}$) | Sampling Best-of-N | REBASE FLOPS ($\times 10^{12}$) | REBASE Best-of-N|
> |-----|-----|----|------|------|
> | 1  | 2.28 | 72.3% | 2.28| 72.3% |
> | 2   | 4.58 |79.7%   | 4.93  |79.6%    |
> | 4    | 9.15  |84%   | 9.92  | 85%   |
> | 8   | 18.3 |86.5%  | 19.8   |87.3%    |
> | 16    |36.6  | 87.5% |39.7  | 89%   |
> | 32    | 73.2  | 88.6%  | 79.4   | 89.1%  |
> | 64   | 146  | 89% | 159  | 89.5% |
> | 128   | 293   | 89.4%  |317 |90%  |
> | 256 | 586 | 89.7% |    |   |
>
> We find that experimental results on Llama3 share similar trends with our original observation. We are grateful for your suggestion, which helps us better demonstrate the robustness of our findings and keep the research up-to-date.
>
> We sincerely hope that our responses address your concerns and you reevaluate our work based on the responses. Thank you again for your time!

---

> > ### Author Response · Authors · 2024-11-29
> > **Looking forward to your valuable feedback!**
> >
> > Dear Reviewer 8MgV,
> >
> > Thank you again for your time and efforts in reviewing our paper. We have carefully responded to each of your questions. Given that the author-reviewer discussion deadline is approaching, we would greatly appreciate it if you could kindly review our responses and share your valuable feedback. If you have any remaining concerns, we would be more than happy to discuss them further.
> >
> > Best regards,
> >
> > Authors of Paper 6864

---

> ### Author Response · Authors · 2024-11-25
>
> Dear Reviewer,
>
> Thank you again for your time and effort in reviewing our paper. If you could spare a moment to review our response and share any further comments, it would be greatly appreciated!
>
> Warm regards,
>
> Authors of Paper 6864

---

### Author Response · Authors · 2024-11-21
**General Response**

We sincerely thank all the reviewers for their insightful comments and valuable feedback.
- We are particularly pleased to see that all the reviewers agree that our paper is well-motivated and well-written.  Specifically, the focused topic (on analyzing inference scaling law with respect to maximizing performance under compute budget) is intriguing (8MgV), well-justified (ExJx), interesting and timely (2jwG, oZkX).
- We also appreciate the recognition by reviewers 8MgV, 2jwG, and oZkX on  the significance of our insights and empirical observations  for the LLM community and practitioners. Scaling inference-time compute with compute-optimal inference strategies could result in huge performance gains, even larger than scaling model size.
- Additionally, we are glad to see that our theoretical analysis is considered well-constructed and consistent with the experimental results ( Reviewers 2jwG and oZkX).
- Reviewers 8MgV, 2jwG, and oZkX also raised a question regarding the generalizability of our results beyond the Mistral and Lemma models, as well as their applicability to non-mathematical tasks.To address this, we have conducted additional experiments using the Llama3-8B-Instruct model on the MATH and GSM8K datasets, as well as the code generation task MBPP. These results further support our conclusions and demonstrate the broader applicability of our findings.

---

### Meta-Review · Area_Chair_848D · 2024-12-20

**Metareview:**

The paper studies scaling laws for _inference_, wherein test behaviour is studied as a function of inference compute budget. The paper studies the role of model size and _inference algorithm_ (e.g., greedy, majority vote) in this behaviour. It is demonstrated that smaller models with more expensive inference algorithms can dominate larger models with simpler algorithms. The paper also proposes a new inference algorithm (REBASE) that yields better tradeoffs than common existing approaches.

Scaling laws are of considerable interest, and the paper considers a practically relevant scenario where there is a budget on test-time compute. The empirical findings are clearly presented, and are largely intuitive; notably, careful choice in inference strategy can overcome differences in model size when it comes to inference compute-optimal prediction.

The original submission had a few limitations, of varying import.

**Relation to prior work**. The paper omits discussion of very relevant recent works.

(1) Sardana et al., "Beyond Chinchilla-Optimal: Accounting for Inference in Language Model Scaling Laws", ICML 2024. In particular:

- Sardana et al. consider Pareto optimal choices of model size and training tokens, based on _both training & inference FLOPs_, and the pretraining loss. They consider a fixed _inference algorithm_.

- the present paper considers Pareto optimal choices of model size and inference algorithm, based on inference FLOPs _only_, and the downstream error. They consider a fixed _training token budget_.

In the AC's reading, the works are complementary, and one could imagine combining both analyses for a more holistic understanding of LM performance. In particular, Sardana et al. does not seem to consider the impact of different sampling strategies (e.g., greedy versus majority vote), which appears to be a primary focus of the paper. Nonetheless, given the topical similarity, it is imperative that the work is cited and discussed.


(2) Snell et al., "Scaling LLM Test-Time Compute Optimally can be More Effective than Scaling Model Parameters", arXiV 2024. This works is more contemporaneous than the previous one -- being made public shortly before the ICLR submission deadline -- but studies a highly similar problem of the benefits of expending more inference time to improve LLM quality. Some of the conclusions are also highly related, e.g., quoting from the paper, _"in some settings it is be more effective to pretrain smaller models with less compute, and then apply test-time compute to improve model outputs"_. Further, some of the empirical analysis in the paper is arguably more fine-grained than the present submission, e.g., breaking down prompts by hardness when studying scaling with inference compute.

**Models**. The original paper primarily studies Pythia and Llemma models, along with the Mistral-7B model. A concern was whether the results generalise to newer models, such as LLaMa. The revision reports results on Llama3-8B-Instruct, which are generally consistent with those in the body. There is still scope for further experimentation here, e.g., scaling up the model size even further.

**Tasks**. As acknowledged by the authors, the paper's focus is on a somewhat narrow range of tasks. This
could however be acceptable for an initial study of this problem. The authors also added results on MBPP, which again are generally consistent with those in the body.

Overall, while the paper has some scope for strengthening, we believe that on balance it could be of value to the community. We would strongly encourage discussion of the related works above, particularly the contemporaneous work of Snell et al.

**Additional Comments On Reviewer Discussion:**

Reviewer opinion was a little divided, with a concern raised around the technical novelty of the paper. Originally, this was around the contribution of the REBASE algorithm, which builds on ideas of reward models and MCTS. In the discussion, a comment arose about a relevant prior work of Sardana et al.

The author response argued the paper's main contribution was around formally introducing and framing the inference scaling law problem. It was argued that the REBASE algorithm is intended only an illustrative tool for the point that the choice of inference algorithm can affect the Pareto optimal behaviour. This argument is found to be reasonable by the AC. However, per above, the relation of the work to Sardana et al. is important to discuss and cite.

---

### Decision · Program_Chairs · 2025-01-22

Accept (Poster)